# OPEN-SET RECOGNITION:
# A GOOD CLOSED-SET CLASSIFIER IS ALL YOU NEED?

**Sagar Vaze**[*]     **Kai Han**[*†]     **Andrea Vedaldi**[*]     **Andrew Zisserman**[*]
[*]Visual Geometry Group, University of Oxford
[†]The University of Hong Kong
{sagar,vedaldi,az}@robots.ox.ac.uk     kaihanx@hku.hk

## ABSTRACT

The ability to identify whether or not a test sample belongs to one of the semantic classes in a classifier's training set is critical to practical deployment of the model. This task is termed open-set recognition (OSR) and has received significant attention in recent years. In this paper, we first demonstrate that the ability of a classifier to make the 'none-of-above' decision is highly correlated with its accuracy on the closed-set classes. We find that this relationship holds across loss objectives and architectures, and further demonstrate the trend both on the standard OSR benchmarks as well as on a large-scale ImageNet evaluation. Second, we use this correlation to boost the performance of the maximum softmax probability OSR 'baseline' by improving its closed-set accuracy, and with this strong baseline achieve state-of-the-art on a number of OSR benchmarks. Similarly, we boost the performance of the existing state-of-the-art method by improving its closed-set accuracy, but the resulting discrepancy with the strong baseline is marginal. Our third contribution is to present the 'Semantic Shift Benchmark' (SSB), which better respects the task of detecting *semantic* novelty, as opposed to low-level distributional shifts as tackled by neighbouring machine learning fields. On this new evaluation, we again demonstrate that there is negligible difference between the strong baseline and the existing state-of-the-art. Code available at: https://github.com/sgvaze/osr_closed_set_all_you_need.

## 1 INTRODUCTION

Given the success of modern deep learning systems on closed-set visual recognition tasks, a natural next challenge is *open-set recognition* (OSR) (Scheirer et al., 2013). In the closed-set setting, a model is tasked with recognizing a set of categories that remain the same during both training and testing phases. In the more realistic open-set setting, a model must not only be able to distinguish between the training classes, but also indicate if an image comes from a class it has not yet encountered.

The OSR problem was initially formalized in (Scheirer et al., 2013) and has since inspired a rich line of research (Bendale & Boult, 2016; Chen et al., 2020a; Ge et al., 2017; Neal et al., 2018; Sun et al., 2020; Zhang et al., 2020; Shu et al., 2020). The standard baseline for OSR is a model trained with the cross-entropy loss on the known classes. At test time, the maximum value of the softmax probability vector is used to decide if an input belongs to the known classes or not. We henceforth refer to this method as the 'baseline' or 'maximum softmax probability (MSP) baseline'. Most existing literature reports significantly outperforming this OSR baseline on standard benchmarks of re-purposed image recognition datasets, including MNIST (LeCun et al., 2010) and TinyImageNet (Le & Yang, 2015).

In this paper we reappraise these approaches, by asking whether a well-trained closed-set classifier can perform as well as recent algorithms, and by analyzing the benchmark datasets. To do this, we first investigate the relationship between the closed-set and open-set performance of a classifier (sec. 3). Though one may expect stronger closed-set classifiers to overfit to the training classes (Recht et al., 2019; Zhang et al., 2017), and so perform poorly for OSR, we show instead that the closed-set and open-set performance are highly correlated. We show this trend holds across datasets, objectives and model architectures, and further demonstrate the trend on an ImageNet-scale evaluation.

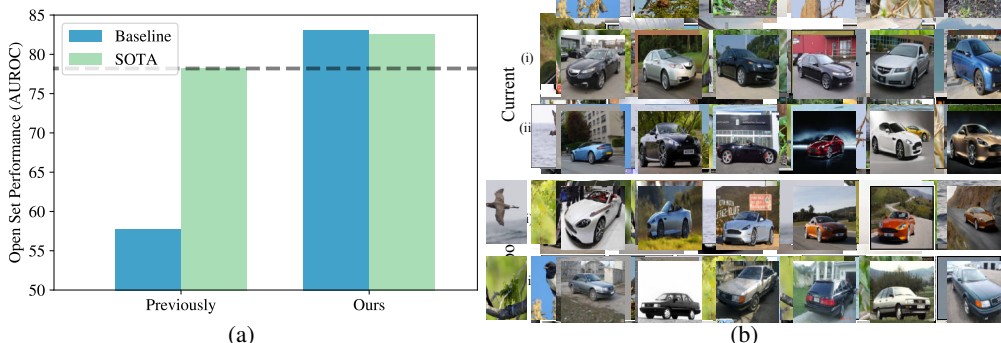

(a)             (b)

Figure 1: (a) We show that we can push OSR baseline performance to be competitive with or surpass state-of-the-art methods (shown, ARPL + CS (Chen et al., 2021)). (b) We propose the 'Semantic Shift Benchmark' datasets for OSR, which are larger scale and give precise definitions of what constitutes a 'new class'.

Secondly, following this observation, we show that the open-set performance of a classifier can be improved by enhancing its closed-set accuracy, tapping the numerous recent advances in image classification (Loshchilov & Hutter, 2017; Szegedy et al., 2016; Cubuk et al., 2020; Bello et al., 2021). Specifically, we introduce strategies such as more augmentation, better learning rate schedules and label smoothing, that significantly improve the closed-set performance of the MSP baseline (sec. 4). We also propose the use of the maximum logit score (MLS), rather than normalized softmax probabilities, as an open-set indicator. With these adjustments, we push the baseline to become competitive with or outperform state-of-the-art OSR methods, substantially outperforming the currently reported baseline figures. Notably, we surpass state-of-the-art figures on four of the six OSR benchmark datasets.

Furthermore, we transfer these improvements to two previous OSR methods, including the current state-of-the-art from (Chen et al., 2021). While this does boost its performance, we observe that there is negligible difference with that of the improved 'MLS' baseline (see fig. 1a). This finding is important because it allows us to better assess recent reported progress in the area.

Finally, we turn to the experimental setting for OSR (sec. 5). Current OSR benchmarks are both small scale and lack a specific definition of what constitutes a 'visual class'. As an alternative, we propose the 'Semantic Shift Benchmark' suite (SSB). We propose the use of fine-grained datasets — including CUB (Wah et al., 2011), Stanford Cars (Krause et al., 2013) and FGVC-Aircraft (Maji et al., 2013) — which all have clear definitions of a semantic class (see fig. 1b), as well as an ImageNet-scale evaluation based on the full ImageNet database (Ridnik et al., 2021). Furthermore, we construct open-set splits with an explicit focus on *semantic novelty*, which we hope better separates this avenue of research from related machine learning sub-fields such as out-of-distribution (Hendrycks & Gimpel, 2017) and anomaly detection (Kwon et al., 2020). Our proposed splits also offer a better way of quantifying open-set difficulty; we find that different splits lead to a much larger discrepancy in open-set performance than the current measure of open-set difficulty 'openness' (Scheirer et al., 2013), which focuses only on the number of open-set classes. We evaluate our strong baseline as well as the state-of-the-art method on this new configuration to encourage future research in this direction.

## 2   RELATED WORK

**Open-set recognition.**   Seminal work in (Scheirer et al., 2013) formalized the task of open-set recognition, and has inspired a number of subsequent works in the field. (Bendale & Boult, 2016) introduced the first deep learning approach for OSR, OpenMax, based on the Extreme Value Theory (EVT). GANs have also been used to tackle the task (Ge et al., 2017; Neal et al., 2018). OSRCI (Neal et al., 2018) generates images similar to those in the training set but that do not belong to any of the known classes, and uses the generated images to train an open-set classifier. This work also established the existing OSR benchmark suite. (Kong & Ramanan, 2021) achieve strong OSR performance by using an adversarially trained discriminator to delineate closed from open-set images, leveraging real open-set images for model selection. Other approaches include reconstruction based methods (Yoshihashi et al., 2019; Oza & Patel, 2019; Sun et al., 2020) which use poor test-time reconstruction as an open-set indicator, and prototype-based methods (Shu et al., 2020; Chen et al.,

2020a; 2021) which represent known classes with learned prototypes, and identify open-set images based on distances to the prototypes.

**State-of-the-art.** In this work, we compare against methods which achieve state-of-the-art in the controlled OSR setting (with no extra data for training or model selection, for instance as demonstrated in (Kong & Ramanan, 2021)). To our knowledge, these methods are ARPL (Adversarial Reciprocal Point Learning) (Chen et al., 2020a; 2021) and OpenHybrid (Zhang et al., 2020), which we detail in sec. 3.1 and sec. 4 respectively. In this paper, we show that the MSP baseline can be competitive with or outperform the more complex methods listed above. Finally, we note recent works (Zhou et al., 2021; Miller et al., 2021; Guo et al., 2021) with which we do not compare as they report lower performance than ARPL and OpenHybrid.

**Related subfields.** OSR is also closely related to out-of-distribution (OoD) detection (Hendrycks & Gimpel, 2017; Liang et al., 2018; Hsu et al., 2020), novelty detection (Abati et al., 2019; Perera et al., 2019; Tack et al., 2020), anomaly detection (Hendrycks et al., 2019; Kwon et al., 2020; Bergman & Hoshen, 2020) and novel category discovery (Han et al., 2019; 2020; 2021). Amongst these, OoD is perhaps the most widely studied and is similar in nature to OSR. As noted by (Dhamija et al., 2018; Boult et al., 2019), OSR is similar to the OoD problem with an additional multi-way classification component between known categories. In fact, there is currently significant overlap in the evaluation datasets between these settings, though cross-setting comparisons are difficult due to different evaluation protocols. Specifically, the OoD setting permits the use of additional data as examples of 'OoD' data during training. (Chen et al., 2021) and (Zhang et al., 2020) evaluate their OSR methods on OoD benchmarks, with both showing competitive results despite not having access to additional data during training. In this paper, we distinguish the OSR problem from OoD and other related fields by proposing a new suite of benchmarks. While OoD encompasses all forms of distributional shift, including those based on low-level features, OSR specifically refers to *semantic* novelty. We propose new benchmarks that respect this distinction.

## 3 Correlation between closed-set and open-set performance

One may expect that stronger closed-set classifiers have overfit their learned representations to the closed-set categories, and thus perform poorly for OSR (Recht et al., 2019; Zhang et al., 2017). Furthermore, existing literature largely considers the closed and open-set tasks separately, with works generally emphasising good open-set performance despite no degradation in closed-set accuracy (Neal et al., 2018; Zhou et al., 2021; Miller et al., 2021). On the contrary, in this section we show that the closed-set and open-set performance of classifiers are strongly correlated. We first demonstrate this for the baseline and a state-of-the-art method on the standard OSR benchmarks (sec. 3.1) and then on a large scale evaluation across a number of model architectures (sec. 3.2).

**Open-set recognition.** We formalize the problem of OSR, and highlight its differences from *closed-set recognition*. First, consider a labelled training set for a classifier $\mathcal{D}_{\text{train}} = \{(\mathbf{x}_i, y_i)\}_{i=1}^{N} \subset \mathcal{X} \times \mathcal{C}$. Here, $\mathcal{X}$ is the input space (*e.g.*, images) and $\mathcal{C}$ is the set of 'known' classes. In the closed-set scenario, the model is evaluated on a test set in which the labels are also drawn from the same set of classes, *i.e.*, $\mathcal{D}_{\text{test-closed}} = \{(\mathbf{x}_i, y_i)\}_{i=1}^{M} \subset \mathcal{X} \times \mathcal{C}$. In the closed-set setting, the model returns a distribution over the known classes as $p(y|\mathbf{x})$. Conversely, in OSR, test images may also come from unseen classes $\mathcal{U}$, giving $\mathcal{D}_{\text{test-open}} = \{(\mathbf{x}_i, y_i)\}_{i=1}^{M'} \subset \mathcal{X} \times (\mathcal{C} \cup \mathcal{U})$. In the open-set setting, in addition to returning the distribution $p(y|\mathbf{x}, y \in \mathcal{C})$ over known classes, the model also returns a score $\mathcal{S}(y \in \mathcal{C}|\mathbf{x})$ to indicate whether or not the test sample belongs to *any* of the known classes.

### 3.1 Baseline and state-of-the-art on standard benchmarks

We first experiment with three representative open-set recognition methods across the standard benchmark datasets in the literature (Neal et al., 2018; Oza & Patel, 2019; Sun et al., 2020; Chen et al., 2020a; Zhang et al., 2020). The methods include the standard MSP baseline as well as two variants of ARPL (Chen et al., 2021). We use the standard network from the open-set literature (Neal et al., 2018), a lightweight model similar to the VGG architecture (Simonyan & Zisserman, 2015) which we henceforth refer to as 'VGG32' (refer to appendix D for details). The three methods are summarised below, followed by a description of the most commonly used benchmarks.

**Methods. Maximum Softmax Probability (MSP, baseline)**: The model is trained for closed-set classification using the cross-entropy loss between a one-hot target vector and the softmax output

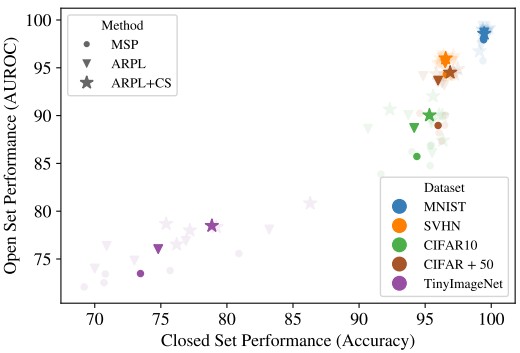

Figure 2: **Correlation between closed set performance (accuracy) and open-set performance (AUROC).** We train three methods on the standard open-set benchmark datasets, including the MSP baseline, ARPL and ARPL + CS (Chen et al., 2021). Foreground points in bold show results averaged across five 'known/unknown' class splits for each method-dataset pair (following standard practise in the OSR literature) while background points, shown feint, indicate results from the underlying individual splits.

$p(y|\mathbf{x})$ of the classifier. This training strategy, along with the use of the maximum softmax probability as $\mathcal{S}(y \in \mathcal{C}|\mathbf{x}) = \max_{y \in \mathcal{C}} p(y|\mathbf{x})$, is widely used in both the OSR and OoD literature as a baseline (Hendrycks & Gimpel, 2017). **ARPL** (Chen et al., 2021): This method is an extension of the recent RPL (Reciprocal Point Learning) optimization strategy (Chen et al., 2020a). Here, the probability that a sample belongs to a class is proportional to its distance from a learned 'reciprocal point' in the feature space. A reciprocal point aims to represent 'otherness' with respect to a class, with the intuition being that open-set examples are different to all known classes. ARPL extends RPL by computing feature distances as the sum of both the Euclidean and cosine distances. In this case, $\mathcal{S}(y \in \mathcal{C}|\mathbf{x})$ is equal to the maximum distance in feature space between the image and any reciprocal point. **ARPL + CS** (Chen et al., 2021) augments ARPL with 'confusing samples': adversarially generated latent points to stand in for 'unseen class' samples. The confusing samples are encouraged to be equidistant from all reciprocal points, with the same open-set scoring rule used as in ARPL. We train both ARPL and ARPL + CS based on the official public implementation (Chen et al., 2021).

**Datasets.** We train the above methods on the standard benchmark datasets for open-set recognition. In all cases, the model is trained on a subset of classes, while other classes are reserved as 'unseen' for evaluation. **MNIST** (LeCun et al., 2010), **SVHN** (Netzer et al., 2011), **CIFAR10** (Krizhevsky, 2009): These are ten-class datasets, with MNIST and SVHN containing images of hand-written digits and street-view house numbers respectively. Meanwhile, CIFAR10 is a generic object recognition dataset containing natural images from ten diverse classes including animals and vehicles. In these cases, the open-set methods are evaluated by training on six classes, while using the other four classes for testing ($|\mathcal{C}| = 6; |\mathcal{U}| = 4$). **CIFAR + N** (Krizhevsky, 2009): In an extension to the CIFAR10 evaluation protocol, open-set algorithms are benchmarked by training on four classes from CIFAR10, while using $N$ classes from CIFAR100 for evaluation, where $N$ denotes either 10 or 50 classes ($|\mathcal{C}| = 4; |\mathcal{U}| \in \{10, 50\}$). **TinyImageNet** (Le & Yang, 2015): In the final and most challenging case, exisiting open-set algorithms are evaluated on the TinyImageNet dataset. This dataset contains 200 classes sub-sampled from ImageNet (Russakovsky et al., 2015), with 20 classes used for training and 180 as unknown ($|\mathcal{C}| = 20; |\mathcal{U}| = 180$).

**Experimental setup.** At test time, the model is fed test images from both known and novel classes, and is tasked with making a binary 'known/unknown' decision on a per-image basis. Following standard practise in the OSR literature, the threshold-free area under the Receiver-Operator curve (AUROC) is used as an evaluation metric. We train with the same hyper-parameters as in (Chen et al., 2021) and, following standard practise, train on five different splits of closed and open-set classes for each dataset and method combination. When evaluating on existing benchmarks throughout this paper, we use the same data splits as (Chen et al., 2021).

**Results.** Fig. 2 gives the AUROC (open-set performance) against the Top-1 multi-way classification accuracy (closed-set performance). We show the averaged results as well as the individual split results, omitting the CIFAR+10 setting for clarity (as the scatter points are almost coincident with the CIFAR+50 setting). It is clear that there is a positive correlation between the closed-set accuracy and open-set performance: we find a Pearson Product-Moment correlation $\rho = 0.95$ between the accuracy and AUROC, indicating a roughly linear relationship between the two metrics.

**Discussion.** To justify our findings theoretically, we look to the model calibration literature (Guo et al., 2017). Intuitively, model calibration aims to quantify whether the model 'knows when it doesn't know', in that low confidence predictions are correlated with high error rates. Specifically, assume a classifier, $f(\mathbf{x})$, returns probabilities for each class, making predictions as $\hat{y} = \arg\max f(\mathbf{x})$.

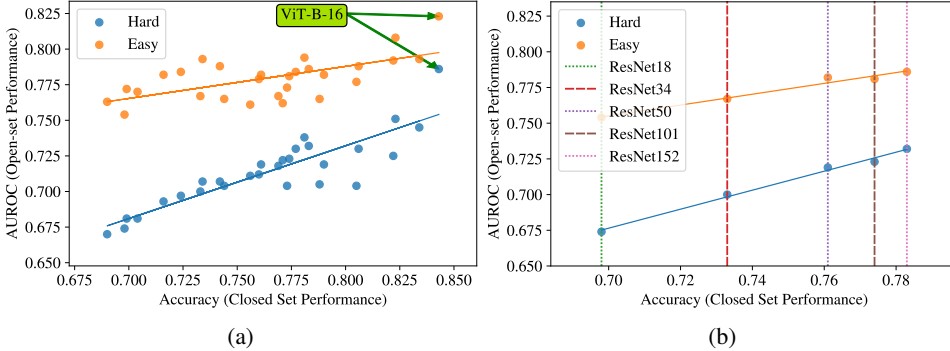

Figure 3: (a) Open-set results on a range of architectures on the ImageNet dataset. 'Easy' and 'Hard' OSR splits are constructed from the ImageNet-21K-P dataset. (b) ImageNet open-set results within a single model family (ResNet).

Further assume labelled input-output pairs, $(\mathbf{x}, y) \subset \mathcal{X} \times \mathcal{C}$, where $\mathcal{C}$ is the label space. Then, the classifier is said to be perfectly calibrated if:

$$P(\hat{y} = y | f(x) = p) = p \quad \forall p \in [0, 1] \tag{1}$$

It is further true that if a classifier is trained with a *proper scoring rule* (Gneiting et al., 2007) on infinite data, then the classifier will be perfectly calibrated at the loss function's minimum (Minderer et al., 2021). Many losses used to train deep networks are proper scoring rules (*e.g.*, the cross-entropy loss). Thus, assuming that generalization error on the test set is correlated with the infinite-data loss value, we would suspect models with lower generalization (test) error to be better calibrated. If we use low-confidence predictions as an indicator that a test sample belongs to a new semantic class, we would expect stronger models to be better open-set detectors.

## 3.2 LARGE-SCALE EXPERIMENTS AND ARCHITECTURE ABLATION

So far, we have demonstrated the correlation between closed and open-set performance on a single, lightweight architecture and on small scale datasets – though we highlight that they are the standard existing benchmarks in the OSR literature. Here, we experiment with a range of architectures on a large-scale dataset (ImageNet).

**Methods.** We experiment with architectures from a number of popular model families, including VGG (Simonyan & Zisserman, 2015), ResNet (He et al., 2016) and EfficientNet (Tan & Le, 2019). We further include results for the recently proposed non-convolutional ViT (Dosovitskiy et al., 2021) and MLP-Mixer (Tolstikhin et al., 2021; Melas-Kyriazi, 2021) models. All models were trained with the cross-entropy objective for classification.

**Dataset.** For large-scale evaluation, we leverage the recently released ImageNet-21K-P (Ridnik et al., 2021). This dataset contains a subset of the full ImageNet database, processed and standardized to remove small classes and leaving around 11K object categories. Note that ImageNet-21K-P is a strict superset of ImageNet-1K (ILSVRC12). As such, models are trained on the standard 1000 classes from ImageNet-1K, and we select two 1000-category subsets from the *disjoint* categories in ImageNet-21K-P as the open sets. Differently to existing practise on the standard datasets, our two open-set splits for ImageNet are not randomly sampled, but rather designed to be 'Easy' and 'Hard' based on the semantic similarity of the open-set categories to the training classes. In this way we better capture a model's ability to identify *semantic* novelty as opposed to low-level distributional shift. This idea and split construction details are expanded upon in sec. 5. For both 'Easy' and 'Hard' splits, we have $|\mathcal{C}| = 1000$ and $|\mathcal{U}| = 1000$.

**Results.** Fig. 3a shows our open-set results on ImageNet. Once again, we find a positive correlation between closed and open-set performance. In this case we find the linear relationship to be weaker, with $\rho = 0.88$ for the 'Hard' evaluation and $\rho = 0.63$ for the 'Easy'. This is unsurprising given the large discrepancy in architecture styles. In general, we do not find any particular model family to be remarkably better for OSR than others. The exception is the ViT model (highlighted), which bucks the OSR trend for both 'Easy' and 'Hard' splits. When looking within a single model family, we

find the linear relationship to be substantially strengthened. Fig. 3b demonstrates the trend within the ResNet family, with $\rho = 1.00$ and $\rho = 0.99$ for the 'Easy' and 'Hard' OSR splits respectively.

**Discussion.** We again note that the ViT model, despite its size (86M parameters) and few inductive biases (no convolutions), does not overfit its representation to the training classes. The fact that it outperforms the OSR trend supports recent findings on the benefits of purely attention-based vision models (including similar findings in (Fort et al., 2021)), as well as the benefits of good closed-set performance for OSR. Finally, we note the practical utility of our findings in sec. 3. Namely, the fact that the open and closed-set performance are correlated allows OSR to readily improve with the extensive research in standard image recognition.

## 4   A GOOD CLOSED-SET CLASSIFIER IS ALL YOU NEED?

In this section, we demonstrate that we can leverage the correlation established in sec. 3 to improve the performance of the baseline OSR method. Specifically, we improve the closed-set accuracy of the maximum softmax probability (MSP) baseline and, in doing so, make it competitive with or stronger than state-of-the-art open-set models. Specifically, we achieve new state-of-the-art figures on four of the six OSR benchmarks.

We find that we can significantly improve the MSP baseline performance by leveraging techniques from the image recognition literature, such as longer training, better augmentations (Cubuk et al., 2020) and label smoothing (Szegedy et al., 2016). Fig. 4 shows how open-set performance of the baseline model increases as we introduce these changes on the Tiny-ImageNet benchmark. For example: longer training (scatter point 7 - scatter point 8); better augmentations (3 - 5); and ensembling (8 - 9). Full details and a tabular breakdown of the methods used to increase closed-set performance can be found in appendix C.

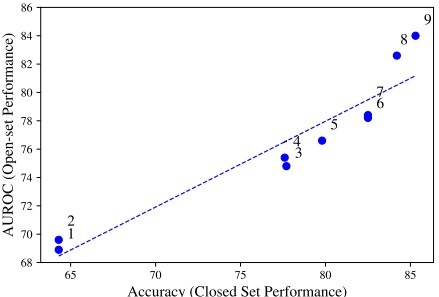

Figure 4: **Gains in open-set performance as closed-set performance increases on TinyImageNet.**

We take these improved training strategies and train the VGG32 backbone on the standard benchmark datasets. We train all models for 600 epochs with a batch size of 128, training models on a single NVIDIA Titan X GPU. We do not include ensemble results for fair comparison with previous methods. Full training strategies and implementation details can be found in  appendices C and D. We report our results as 'Baseline (MSP+)' in table 1.

**Logit scoring rule.** Next, we also change the open-set scoring rule. Previous work has noted that open-set examples tend to have lower feature norms than closed-set ones  (Dhamija et al., 2018; Chen et al., 2021). As such, we propose the use of the maximum *logit* score (MLS) for the open-set scoring rule. Logits are the raw outputs of the final linear layer in a deep classifier, before the softmax operation normalizes these such that the outputs can be interpreted as a probability vector summing to one. As the softmax operation normalizes out much of the feature magnitude information present in the logits, we find logits lead to better open-set detection results. We provide a detailed analysis and discussion of this effect in appendix B. We further provide a more general study of the representations learned with cross-entropy models, including visualizations of the learned feature space. We present results of our maximum logit score baseline as 'Baseline (MLS)' in table 1.

We compare against OpenHybrid (Zhang et al., 2020) and ARPL + CS (Chen et al., 2021), which hold state-of-the-art performances on the standard datasets in the controlled setting (with no extra data for training or model selection). We also compare against OSRCI (Neal et al., 2018), which established the current OSR benchmark suite. While OSRCI and ARPL + CS have been described in sec. 2 and 3.1 respectively, OpenHybrid tackles the open-set task by training a flow-based density estimator on top of the classifier's feature representation, jointly training both the encoder and density model. In this way, a distribution over the training data $\log p(\mathbf{x})$ is learned, which is used to directly provide $\mathcal{S}(y \in \mathcal{C}|\mathbf{x})$. Comparisons with more methods can be found in appendix E.

We find that our MLS baseline substantially improves the previously reported baseline figures, with an average absolute increase in AUROC of 15.6% across the datasets. In fact, MLS surpasses the existing state-of-the-art on the SVHN, CIFAR+10, CIFAR+50 and TinyImageNet benchmarks and is, on average, 0.7% better across the entire suite.

Table 1: **Comparisons of our improved baselines (MSP+, MLS) against state-of-the-art methods on the standard OSR benchmark datasets.** All results indicate the area under the Receiver-Operator curve (AUROC) averaged over five 'known/unknown' class splits. '+' indicates prior methods augmented with improved closed-set optimization strategies, including: MSP+ (Neal et al., 2018), OSRCI+ (Neal et al., 2018) and (ARPL + CS)+ (Chen et al., 2021).

| Method | MNIST | SVHN | CIFAR10 | CIFAR + 10 | CIFAR + 50 | TinyImageNet |
|---|---|---|---|---|---|---|
| Baseline (MSP) (Neal et al., 2018) | 97.8 | 88.6 | 67.7 | 81.6 | 80.5 | 57.7 |
| OSRCI (Neal et al., 2018) | 98.8 | 91.0 | 69.9 | 83.8 | 82.7 | 58.6 |
| OpenHybrid (Zhang et al., 2020) | 99.5 | 94.7 | **95.0** | 96.2 | 95.5 | 79.3 |
| ARPL + CS (Chen et al., 2021) | **99.7** | 96.7 | 91.0 | 97.1 | 95.1 | 78.2 |
| OSRCI+ | 98.5 (-0.3) | 89.9 (-1.1) | 87.2 (**+17.3**) | 91.1 (**+7.3**) | 90.3 (**+7.6**) | 62.6 (**+4.0**) |
| (ARPL + CS)+ | 99.2 (-0.5) | 96.8 (**+0.1**) | 93.9 (**+2.9**) | **98.1** (**+1.0**) | **96.7** (**+1.6**) | 82.5 (**+4.3**) |
| Baseline (MSP+) | 98.6 (**+0.8**) | 96.0 (**+7.4**) | 90.1 (**+22.4**) | 95.6 (**+14.0**) | 94.0 (**+13.5**) | 82.7 (**+25.0**) |
| Baseline (MLS) | 99.3 (**+1.5**) | **97.1** (**+8.5**) | 93.6 (**+25.9**) | 97.9 (**+16.3**) | 96.5 (**+16.0**) | **83.0** (**+25.3**) |

We also take the OSRCI and ARPL + CS algorithms (Neal et al., 2018; Chen et al., 2021), and augment them with our proposed training strategies for a fair comparison, reporting the results under OSRCI+ and (ARPL + CS)+. Specifically, we train them for longer, include label smoothing and use better data augmentations (see appendix D for full details). We also trained OpenHybrid in this controlled setting, but significantly underperformed the reported performance. This is likely because the method was trained for 10k epochs and with a batch size of 1024, which are both $10\times$ larger than those used in these experiments. Note that, despite this, the stronger baseline still outperforms OpenHybrid in a number of cases.

In almost all cases we are able to boost the open-set performance of OSRCI and ARPL+CS, especially for the former. In the case of (ARPL+CS)+, we achieve new state-of-the-art results on the CIFAR+10 and CIFAR+50 benchmarks, and also report a 4.3% boost on TinyImageNet. However, we note that on average, (ARPL+CS)+ is almost indistinguishable from the improved MLS baseline (with 0.03% difference in average open-set performance).

**Discussion.** A number of increasingly sophisticated methods have been proposed for OSR in recent years. Typically, proposed methods have carefully tuned training strategies and hyper-parameters, such as custom learning rate schedules (Zhang et al., 2020), non-standard backbones (Guo et al., 2021) and novel data augmentations (Zhou et al., 2021). Meanwhile, the closed-set accuracy of the methods is often unreported. As such, it is difficult to delineate what proportion of the open-set performance gains come from increases in closed-set accuracy. Our findings in this section suggest that many of the gains could equally be realised through the standard baseline. Indeed, in sec. 5, we propose new evaluation protocols and find that once the closed-set accuracy of ARPL and the baseline are made comparable, there is negligible difference in open-set performance. We further experiment on OoD benchmarks in appendix F and report similarly improved baseline performance.

## 5 SEMANTIC SHIFT BENCHMARK

Current OSR benchmarks have two drawbacks: (1) they all involve small scale datasets; (2) they lack a clear definition of what constitutes a 'semantic class'. The latter is important to delineate the open-set field from other research questions such as out-of-distribution detection (Hendrycks & Gimpel, 2017) and anomaly detection (Kwon et al., 2020). Specifically, OSR aims to identify whether a test image is *semantically* different to the training classes, not whether, for example, the model is uncertain about its prediction or whether there has been a low-level distributional shift.

To address these issues, we propose a new suite of evaluation benchmarks. In this section, we first detail a large-scale ImageNet evaluation (introduced in sec. 3.2) before proposing three evaluations on fine-grained datasets which have clear definitions of a semantic class. Differently to previous work, our evaluation settings all aim to explicitly capture the notion of *semantic novelty*. Finally, we benchmark MLS and ARPL on the new benchmark suite to motivate future research.

### 5.1 PROPOSED BENCHMARK DATATSETS

**ImageNet.** We introduce a large-scale evaluation for category shift, with open-set splits based on semantic distances to the training set. Specifically, we designate the original ImageNet-1K classes for the closed-set, and choose open-set classes from the *disjoint* set of ImageNet-21K-P (Ridnik et al., 2021). We exploit the hierarchical, tree-like semantic structure of the ImageNet database. For

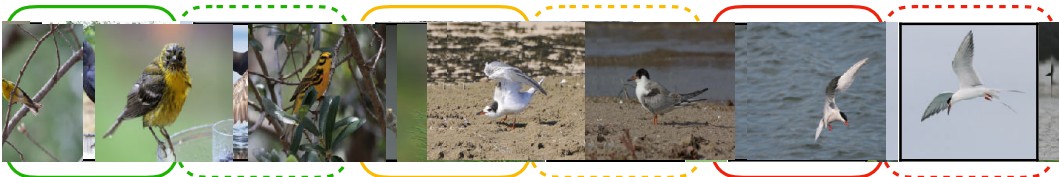

Figure 5: **Open-set class pairs for CUB.** For three difficulties {'Easy' (green/left), 'Medium' (orange/middle), 'Hard' (red/right)}, we show an image from an open-set class (right) and its most similar closed-set class (left). Note that the harder the difficulty, the more visual features (*e.g.*, foot colour or bill shape) the open-set class has in common with the closed-set. Further examples can be found in appendix H.

instance, the class 'elephant' can be labelled at multiple levels of semantic abstraction ('elephant', 'placental', 'mammal', 'vertebrate', 'animal'). Thus, for each pair of classes between ImageNet-1K and ImageNet-21K-P, we define the semantic distance between two classes as the total path distance between their nodes in the semantic tree. We then approximate the total semantic distance from the ImageNet-21K-P classes to the closed-set by summing distances to all ImageNet-1K classes. Finally, we select 'Easy' and 'Hard' open-set splits by sorting the total distances to the closed-set and selecting two sets of 1000 categories. We note that the larger ImageNet database has been used for OSR research previously (Bendale & Boult, 2016; Kumar et al., 2021; Hendrycks et al., 2021). However, we structure explicitly for semantic similarity with ImageNet-1K similarly to concurrent work in (Sariyildiz et al., 2021).

**Fine-grained classification datasets.** Consider the properties of fine-grained visual categorization (FGVC) datasets. These datasets are defined by an 'entry level' category, such as flowers (Nilsback & Zisserman, 2008) or birds (Wah et al., 2011). Within the dataset, all classes are variants of that single category, defining a single *axis of semantic variation*, *e.g.*, 'bird species' in the case of birds. Because the axis of variation is well defined, it is reasonable to expect a classifier to learn it given a number of example classes — namely, to learn what bird species are and how they can be distinguished.

Contrast FGVC datasets with the current OSR benchmarks, such as the CIFAR+10 evaluation. In this case, a model is trained on four CIFAR10 classes such as {*airplane, automobile, ship, truck*}, all of which could be considered 'entry level', before having to identify images from CIFAR100 classes such as {*bicycle, bee, porcupine, baby*} as belonging to new classes. In this case, the axis of variation is much less specific, and it is uncertain whether the OSR model is responding to a true semantic signal or simply to low-level distributional shifts in the 'unseen' data. Furthermore, because of the small number of training classes in the current benchmark settings, it is unrealistic for a classifier to learn such high-level class definitions. We give an illustrative example of this in appendix G.

As a result, we propose three FGVC datasets for OSR evaluation: Caltech-UCSD Birds (CUB) (Wah et al., 2011), Stanford Cars (Krause et al., 2013) FGVC-Aircraft (Maji et al., 2013). These datasets come with labelled attributes (*e.g.*, has_bill_shape::hooked in CUB), which can be used to characterize the differences between classes and thus the degree of semantic shift. We use attributes to construct open-set FGVC class splits which are binned into 'Easy', 'Medium' and 'Hard' classes, with the difficulty depending on the similarity of labelled visual attributes with any of the training classes. We sketch the split-construction process for CUB here, and refer to appendix H for more details on Stanford Cars and FGVC-Aircraft.

Every image in CUB is labelled for the presence of 312 visual attributes such as has_bill_shape::hooked and has_breast_color::yellow. This information is aggregated for each class, resulting in a matrix $M \in [0,1]^{C \times A}$, describing the frequency with which each attribute appears in each class.

Table 2: **Statistics of the Semantic Shift Benchmark.** We show '#Classes(#Test Images)' for the known classes, and for the 'Easy', 'Medium' and 'Hard' open-set classes.

| Dataset | Known | Easy | Medium | Hard |
|---|---|---|---|---|
| CUB | 100 (2884) | 32 (915) | 34 (1004) | 34 (991) |
| Stanford Cars | 98 (3948) | 76 (3170) | - | 22 (923) |
| FGVC-Aircraft | 50 (1668) | 20 (667) | 17 (565) | 13 (433) |
| ImageNet | 1000 (50000) | 1000 (50000) | - | 1000 (50000) |

Treating each row in $M$ as a semantic class descriptor, this allows us to compute the semantic similarity of every pair of classes and, given a set of closed-set classes, identify which remaining classes are 'Easy', 'Medium' and 'Hard' (least to most similar) with respect to the closed-set. Examples of 'Easy', 'Medium' and 'Hard' open-set classes, along with their closest class in the closed-set, are shown in fig. 5 for CUB.

We note that fine-grained OSR has been demonstrated in (Chen et al., 2021; 2020a) on a dataset of 300 aircraft classes. However, this dataset does not come with labelled attributes, making it harder to construct open-set splits with varying levels of semantic similarity to the training set, which is our focus here. Finally, while prior works have recognised the difficulty of OoD detection for more fine-grained data (Bodesheim et al., 2015; Perera & Patel, 2019; Lee et al., 2018a), we propose them for OSR because of their clear definition of a semantic class rather than their increased difficulty. A further discussion of these ideas is presented in appendix G. We provide statistics of the splits from all proposed datasets in table 2, and the splits themselves in the supplementary material.

## 5.2 BENCHMARKING FOR OPEN-SET RECOGNITION

**Evaluation Protocol.** For the 'known/unknown' class decision, we report AUROC as is standard practise, as well as accuracy to allow potential gains in open-set performance to be contextualized in the closed-set accuracy of a model. We also report Open-Set Classification Rate (OSCR) (Dhamija et al., 2018) which measures the trade-off between accuracy and open-set detection rate as a threshold on the confidence of the predicted class is varied. We report results on 'Easy' and 'Hard' splits for all datasets, combining 'Medium' and 'Hard' examples into a single bin when applicable.

In fine-grained classification, it is standard to pre-train models on ImageNet. This is unsuitable for the proposed fine-grained OSR setting, as ImageNet contains overlapping classes with the proposed datasets. Instead, we pre-train the network on Places (Zhou et al., 2017) using MoCoV2 self-supervised weights (Chen et al., 2020b; Zhao et al., 2021). For the ImageNet benchmark, we can train with labels on the ImageNet-1K dataset and evaluate on the unseen classes. We finetune the ARPL model from a pre-trained ImageNet checkpoint.

**Results.** In table 3 we test MLS and ARPL+ (Chen et al., 2021) using a ResNet50 backbone on the proposed benchmarks (we found ARPL + CS to be prohibitively expensive to train in this setting, see appendix D for details). The results corroborate the trends found in sec. 4: strong closed-set classifiers produce open-set results with good AUROC performance, and the MLS baseline performs comparably to the state-of-the-art method. In fact, while we find ARPL+ achieves slightly better AUROC on the ImageNet benchmark, MLS outperforms in terms of OSCR across the board.

Finally, more careful consideration of the semantics of the open-set classes leads to harder splits significantly reducing OSR performance. This is in contrast to 'openness' (Scheirer et al., 2013), the current measure used to assess the difficulty of an OSR problem, dependent on the ratio of the number of closed to open-set classes. For instance, in the ImageNet case, we find the harder split to be lead to around 6% worse AUROC for both methods. We also experimented with randomly subsampling first 1K and then 10K open-set classes, finding that introducing more classes during evaluation only reduced open-set performance by around 0.6% (10× less than our proposed splits).

Table 3: **OSR results on the Semantic Shift Benchmark.** We measure the closed-set classification accuracy and AUROC on the binary open-set decision. We also report OSCR, which measures the trade-off between open and closed-set performance. OSR results are shown on 'Easy / Hard' splits.

| Method | CUB | | | SCars | | | FGVC-Aircraft | | | ImageNet | | |
|---|---|---|---|---|---|---|---|---|---|---|---|---|
| | Acc. | AUROC | OSCR | Acc. | AUROC | OSCR | Acc. | AUROC | OSCR | Acc. | AUROC | OSCR |
| ARPL+ | 85.9 | 83.5 / 75.5 | 76.0 / 69.6 | 96.9 | **94.8 / 83.6** | **92.8 / 82.3** | 91.5 | 87.0 / 77.7 | 83.3 / 74.9 | 78.1 | **79.0 / 73.6** | 65.9 / 62.6 |
| MLS | **86.2** | **88.3 / 79.3** | **79.8 / 73.1** | **97.1** | 94.0 / 82.2 | 92.2 / 81.1 | **91.7** | **90.7 / 82.3** | **86.8 / 79.8** | **78.8** | 78.2 / 72.6 | **66.1 / 62.7** |

## 6 CONCLUSION

In this work we have demonstrated a strong correlation between the closed-set and open-set performance of models for the task of open-set recognition. Leveraging this finding, we have demonstrated that a well-trained closed-set classifier, using the maximum logit score (MLS) at test-time, can be competitive with or outperform existing state-of-the-art methods. Though we believe OSR is a critical problem which requires further investigation, our findings give us insufficient evidence to reject our titular question of 'is a good closed-set classifier all you need?'. We have also proposed the 'Semantic Shift Benchmark' suite, which isolates *semantic* shift from other low-level distributional shifts. Our proposed benchmark suite allows controlled study of semantic novelty, including stratification of the degree of semantic shift.

## ACKNOWLEDGEMENTS

We would like to thank Andrew Brown for many interesting discussions on this work. This research is funded by a Facebook AI Research Scholarship, a Royal Society Research Professorship, and the EPSRC Programme Grant VisualAI EP/T028572/1.

## ETHICS STATEMENT

Open-set recognition is of immediate relevance to the safe and ethical deployment of machine learning models. In real-world settings, it is unrealistic to expect that all categories of interest to the user will be represented in the training set. For instance, in an autonomous driving scenario, forcing the model to identify every object as an instance of a training category could lead it to make unsafe decisions.

When considering potential negative societal impacts of this work, we identify the possibility that OSR research may lead to complacent consideration of the training data. As we have demonstrated, OSR models are far from perfect and cannot be exclusively relied upon in practical deployment. As such, it remains of critical importance to carefully curate training data and ensure its distribution is representative of the target task.

Finally, we comment on the dataset privacy considerations for the existing and proposed benchmarks. All datasets are licensed for academic/non-commercial research. However, CIFAR, TinyImageNet and ImageNet contain some personal data for which consent was likely not obtained. The proposed FGVC datasets have the added benefit of containing no personal information.

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

## A EXPANSION OF FIG. 2 OF THE MAIN PAPER WITH STANDARD DEVIATIONS

For completeness, we include another version of fig. 2 which includes OSRCI models (Neal et al., 2018) in fig. 6. We find the correlation between the closed and open-set performance continues to hold with the inclusion of this additional method. We further report the standard deviations of this plot in table 4. It can be seen that, for the same dataset, the standard deviations of all four methods appear to be similar. The standard deviations on the most challenging TinyImageNet benchmark is greater than on the other datasets.

Finally, we note in fig. 6 that the trend seems less clear at very high accuracies. This may be because AUROC also becomes very high, making it difficult to identify clear patterns. However, it may also indicate that the relationship between the metrics becomes weaker as closed-set performance saturates.

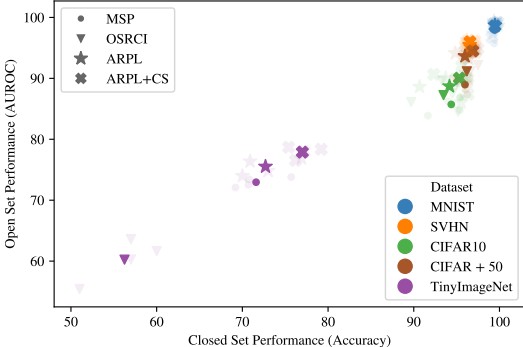

Figure 6: **Correlation between open-set and closed-set performances on the standard OSR benchmarks.** This plot is similar to fig. 2 but includes scatter points for OSRCI (Neal et al., 2018).

Table 4: **Standard deviations of our experiments in fig. 2 of the main paper.** We report the standard deviations for both the closed-set and open-set performance (accuracy/AUROC) across the five 'known/unknown' class splits.

| Method | MNIST | SVHN | CIFAR10 | CIFAR + 50 | TinyImageNet |
|---|---|---|---|---|---|
| MSP | 0.20/1.29 | 0.36/0.55 | 1.64/1.34 | 0.79/1.23 | 4.83/1.36 |
| OSRCI | 0.22/0.52 | 0.47/2.97 | 1.99/1.80 | 0.63/1.47 | 3.27/3.02 |
| ARPL | 0.21/0.77 | 0.43/0.79 | 2.10/1.56 | 0.66/0.44 | 5.40/1.63 |
| ARPL + CS | 0.29/1.04 | 0.51/0.31 | 1.70/1.68 | 0.63/0.23 | 4.40/1.55 |

## B ANALYSING THE CLOSED-SET AND OPEN-SET CORRELATION

Here, we aim to understand why improving the closed-set accuracy may lead to increased open-set performance through the MSL baseline. To this end, we train the VGG32 model on the CIFAR10 benchmark setting with the cross-entropy loss. We train the model both with a feature dimension of $D = 128$ (as is standard for this model) as well as with $D = 2$ for feature space visualization. We also train without a bias in the linear classifier for more interpretable features and classification boundaries (so class boundaries radiate from the origin of the feature space). Specifically, we train a model to make predictions as $\hat{\mathbf{y}}_i = \text{softmax}(\mathbf{W}\Phi_\theta(\mathbf{x}_i))$, where $\Phi_\theta(\cdot)$ is a CNN embedding function ($\Phi_\theta(\mathbf{x}) \in \mathbb{R}^D$) and $\mathbf{W} \in \mathbb{R}^{C \times D}$ is the linear classification matrix. Here $C = |\mathcal{C}| = 6$ and $D \in \{2, 128\}$, and we optimise the loss with a one-hot target vector $\mathbf{y}_i$ and batch size $B$, as $-\frac{1}{B}\sum_{i=1}^{B} \mathbf{y}_i \cdot \log(\hat{\mathbf{y}}_i)$.

Next, we interrogate the learned embeddings by plotting the mean vector norm of the features from all test images, for both the known and unknown classes, as training proceeds. These are shown in fig. 7a and fig. 7b for the models with $D = 128$ and $D = 2$ respectively. We also show the average vector norm for the per-class weights in the linear classifiers as dashed lines. Furthermore, snapshots of how these images are embedded for the model with $D = 2$ are shown in fig. 7d to 7f at representative epochs. The plots of the mean feature norms show that, at the start of training, all

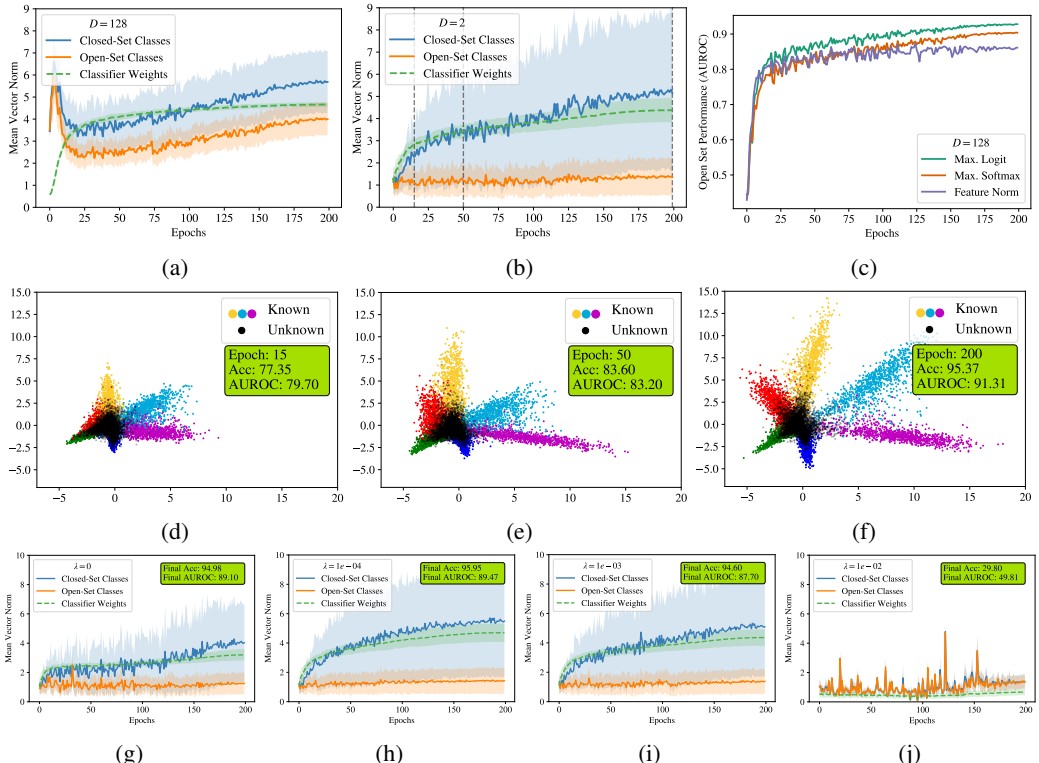

Figure 7: **Plots showing how the feature representations and linear classification weights of a deep classifier evolve as training proceeds (CIFAR10 OSR setting).** (a), (b) show the average feature norm for seen and unseen classes, as well as the per-class vector norms for the weights in the linear classification head, for models with $D = 128$ and $D = 2$ respectively. (c) shows how the open-set performance of the classifier with $D = 128$ develops as training proceeds, using three different OSR scoring rules. (d), (e), (f) show the feature projections for images from seen and unseen classes at different epochs (indicated by vertical dashed lines in (b)) for the model with $D = 2$. We show test images from known classes in colour and unknown classes in black. (g) (h) (i) show how classifier weight and feature norms change as a function of weight decay strength ($\lambda$)

images are embedded with a similar magnitude. However, as training proceeds, the magnitude of features for the known classes increases substantially more than for the unknown classes.

To understand this, consider the cross-entropy loss for a single sample in the batch, shown in eq. (2):

$$\mathcal{L}_i(\theta, \mathbf{W}) = -\hat{y}_{i,c} + \log(\sum_{j=1}^{C} \exp(\hat{y}_{i,j})) = -\mathbf{w}_c \cdot \Phi_\theta(\mathbf{x}_i) + \log(\sum_{j=1}^{C} \exp(\mathbf{w}_j \cdot \Phi_\theta(\mathbf{x}_i))) \qquad (2)$$

where $c$ refers to the correct class index, and $\mathbf{w}_j$ refers to the classification vector corresponding to the $j^{th}$ class. Empirically, we find that the linear classifier's weights and the feature norms for known classes increase during training, which is justified as increasing both $|\mathbf{w}_c|$ and $|\Phi_\theta(\mathbf{x}_i)|$ reduces the loss value. Note that we observe this despite training with weight decay, which we omit from eq. (2) for clarity. [1] However, for 'hard' or 'uncertain' training examples (for which the classifier's prediction may be incorrect) the model is encouraged to reduce $\mathbf{w}_j \cdot \Phi_\theta(\mathbf{x}_i) \ \forall j \neq c$ through the second term of eq. (2). While the only way to do this for the $D = 2$ case is to reduce the feature norm (fig. 7b and fig. 7d to 7f), we show in fig. 7a that this also holds true for the $D = 128$ case in which $D > C$. The tendency of deep networks to map 'hard' samples closer to the origin has been noted in (Ranjan et al., 2017).

This suggests that stronger cross-entropy models project features further from the origin, while still ensuring that any 'uncertain' samples have lower feature norms. This, in turn, suggests stronger cross-entropy classifiers would perform better for OSR, with images from novel categories likely to

---

[1]Ablations for different weight decay values are shown in fig. 7g to 7j (we use $\lambda = 1e - 4$ in this paper).

be interpreted as 'uncertain' during evaluation. Our analysis also suggests that cross-entropy training already provides a strong signal and thus a strong baseline for open-set recognition.

Finally, this motivates us to propose the maximum logit score (MLS) to provide our open-set score, *i.e.*, $\mathcal{S}(y \in \mathcal{C}|\mathbf{x}) = \max_{j \in \mathcal{C}} \mathbf{w}_j \cdot \Phi_\theta(\mathbf{x})$, rather than the softmax output as in the standard MSP baseline. Normalizing the logits via the softmax operator cancels out the magnitude information of the feature representation, which we have demonstrated is useful for the OSR decision. Fig. 7c shows how the AUROC evolves as training proceeds when both the maximum logit and maximum softmax value are used for OSR scoring. The plot demonstrates that softmax normalization noticeably reduces the model's ability to make the open-set decision. We also show the OSR performance if we use the feature norm as our open-set score ($\mathcal{S}(y \in \mathcal{C}|\mathbf{x}) = |\Phi_\theta(\mathbf{x})|$), showing that this simple indicator can perform remarkably well.

## C  IMPROVING OPEN-SET PERFORMANCE WITH STRONGER CLOSED-SET CLASSIFIERS

Here, we describe how we improve the open-set performance of the baseline method in sec. 4 of the main paper, and provide a full breakdown of fig. 4. The methods include better learning rate schedules and data augmentations, as well as the use of logits rather than the softmax output for OSR scoring. We document the closed-set and open-set performance on the TinyImageNet dataset (the most challenging of the OSR benchmarks) in table 5. We further include the 'Open Set Classification Rate' (OSCR (Dhamija et al., 2018)) which summarises the trade-off between closed-set accuracy and open-set performance (here, in terms of the False Positive Rate) as the threshold on the open-set score is varied. As demonstrated in sec. 4 of the main paper, the findings of this study generalize well to other datasets.

Table 5: **Breakdown of methods used to improve the closed-set classification accuracy of the baseline method.** All experiments were conducted with a VGG32 backbone over five 'known/unknown' splits of the TinyImageNet dataset. The bracketed number with the Cosine scheduler indicates the number of learning rate restarts used during training. We find a Pearson Product-Moment correlation of **0.93** between the closed-set accuracy and the open-set AUROC.

| | | | Setting | | | | Closed Set (Accuracy) | Open Set (AUROC) | Combined (OSCR) |
|---|---|---|---|---|---|---|---|---|---|
| Epochs | Scheduler | Aug. | Logit Eval | Warmup | Label Smoothing | Ensemble | | | |
| 100 | Step | RandCrop | ✗ | ✗ | ✗ | ✗ | 64.3 | 68.9 | 51.4 |
| 100 | Step | RandCrop | ✓ | ✗ | ✗ | ✗ | 64.3 | 69.6 | 50.7 |
| 200 | Cosine (0) | RandCrop | ✓ | ✗ | ✗ | ✗ | 77.7 | 74.8 | 64.3 |
| 200 | Cosine (0) | CutOut | ✓ | ✗ | ✗ | ✗ | 77.6 | 75.4 | 64.7 |
| 200 | Cosine (0) | RandAug | ✓ | ✗ | ✗ | ✗ | 79.8 | 76.6 | 67.3 |
| 600 | Cosine (2) | RandAug | ✓ | ✗ | ✗ | ✗ | 82.5 | 78.2 | 70.3 |
| 600 | Cosine (2) | RandAug | ✓ | ✓ | ✗ | ✗ | 82.5 | 78.4 | 70.3 |
| 600 | Cosine (2) | RandAug | ✓ | ✓ | ✓ | ✗ | 84.2 | 83.0 | 74.3 |
| 600 | Cosine (2) | RandAug | ✓ | ✓ | ✓ | ✓ | 85.3 | 84.0 | 76.1 |

We first train the baseline with the same hyper-parameters as in (Chen et al., 2021), training for 100 epochs and using a step learning rate schedule, with a basic random crop augmentation strategy. We evaluate using both softmax and logit scoring strategies. It can be seen that using maximum logit scoring gives better open-set performance (AUROC), while softmax scoring appears to be better in terms of OSCR. This is likely due to the fact that softmax normalization cancels the effect of the feature norm, which results in more separable scores that are beneficial to the OSCR calculation.

Here, we are interested in boosting the open-set performance (AUROC) by improving the closed-set accuracy. Hence, we use the maximum logit for open-set scoring as discussed in appendix B. This already gives an open-set performance of $69.6\%$ AUROC, which is significantly higher than the softmax thresholding baseline reported for these datasets in almost all of the comparisons in the literature, which report a baseline $57.7\%$ AUROC. The discrepancy between the reported baseline and our simplest setting is the result of reported figures originating in (Neal et al., 2018), wherein all models were trained only for 30 epochs (according to the publicly shared code) while our simplest model is trained for 100 epochs.

Following this trend, we find that training for longer (200 epochs) and using a better learning rate schedule (cosine annealed schedule (Loshchilov & Hutter, 2017)) significantly enhances both

closed-set and open-set performance. We further find that stronger augmentations boost accuracy, where we leverage RandAugment (Cubuk et al., 2020) to find an optimal strategy. Finally, we find that learning rate warmup and label smoothing (Szegedy et al., 2016) can together significantly increase accuracy. We select the RandAugment and label smoothing hyper-parameters by maximizing closed-set accuracy on a validation set (randomly sampling 20% of the training set).

In summary, we find that simply *leveraging standard training strategies for image recognition models leads to a significant boost in open-set performance*. Specifically, we find that the combination of the above methodologies, including longer training and better augmentations boosts the AUROC to $82.6\%$. Finally, we find that open-set performance can be boosted to $84.0\%$ AUROC by bootstrapping the training data and training $K = 5$ ensembles. The improvements in open-set performance strongly correlate with the boosts to the closed-set accuracy, with $\rho = 0.93$ between accuracy and AUROC.

# D    IMPLEMENTATION DETAILS

## D.1    VGG32 ARCHITECTURE

This backbone architecture is commonly used in the open-set literature (Neal et al., 2018). The model consists of a simple series of nine $3\times3$ convolution layers, with downsampling occurring through strided convolutions every third layer. Batch normalization and LeakyRelu (slope of 0.2) are used after every convolution layer, with dropout used on the input image, and then after the third and sixth layer. Finally, after the ninth layer, the spatial feature is reduced with average pooling to a feature vector with dimensionality $D = 128$. This is fed to the linear classifier (fully connected layer) to give the output logits.

## D.2    STANDARD DATASETS

Here, we describe the experimental setup for our results in sec. 4 of the main paper.

All models were trained on a single 12GB GPU (mostly a NVIDIA Titan X). When optimizing with the cross-entropy loss, training took between 2 and 6 hours for a single class split, depending on the dataset (for instance, training on TinyImageNet took 2.5 hours). All hyper-parameters were tuned on a validation set which was constructed by holding out a randomly sampled 20% of the closed-set training data from a single split of seen/unseen classes.

**Baselines, MSP+/MLS**    We trained the VGG32 model with a batch size of 128 for 600 epochs. For each dataset, we train on five splits of 'known/unknown' classes as is standard practise, training each run with the random seed '0'. We use an initial learning rate of 0.1 for all datasets except TinyImageNet, for which we use 0.01. We train with a cosine annealed learning rate, restarting the learning rate to the initial value at epochs 200 and 400. Furthermore, we 'warm up' the learning rate by linearly increasing it from 0 to the 'initial value' at epoch 20.

We use RandAugment for all experiments, tuning its hyper-parameters on a validation set from a single class split for each dataset. We follow a similar procedure for the label smoothing value $s$, though we find the optimal value to be $s = 0$ for all datasets except TinyImageNet, where it helps significantly at $s = 0.9$.

**(ARPL + CS)+**    We use the same experimental procedure for ARPL + CS (Chen et al., 2021) as for the baselines, again tuning the RandAugment and label smoothing hyperparameters for this method. Here, following the original implementation, we find a batch size of 64 and learning rate of 0.001 lead to better performance on TinyImageNet. This method also took significantly longer to train, taking 7.5 hours per class split on TinyImageNet.

**OSRCI+**    OSRCI involves multiple stages of training, including first training a GAN to synthesize images similar to the training data, before using generated images as 'open-set' examples to train a $(K + 1)$-way classifier (Neal et al., 2018). As our focus is on the effect of improving classification accuracy on open-set performance, we augment the training of the latter stage of OSRCI. We again train the $(K + 1)$-way classifier for 600 epochs with a cosine annealed learning rate schedule and RandAugment. For this method, we find that reducing all learning rates by a factor 10 compared to the baselines significantly improved performance.

## D.3 Proposed Benchmarks

Here, we describe the experimental setup for our results in sec. 5 of the main paper.

**ImageNet.** For this evaluation, we leverage a ResNet50 model pre-trained with the cross-entropy loss on ImageNet-1K from (Wightman, 2019). We evaluate the model directly for our **MLS** baseline. For **ARPL+**, we finetune the pre-trained model for 10 epochs with the ARPL optimization strategy.

**FGVC datasets.** We use a similar experimental setting for the FGVC datasets as we do for the standard benchmarks. Specifically, for both **MSP+/MLS** and **ARPL+**, we again train for 600 epochs, using a cosine annealed learning rate and learning rate warmup. We also re-tune the RandAugment and label smoothing hyper-parameters on a validation set. Differently, however, we use a ResNet50 backbone with $448 \times 448$ image size as is standard in the FGVC literature. We further initialize the network with weights from MoCoV2 training on Places, using an initial learning rate of 0.001 and a batch size of 32. Training for both methods took between one and two days depending on the dataset.

**Note:** We attempted to train ARPL+CS on our proposed datasets but found it computationally infeasible. Specifically, the memory intensive nature of the method meant we could only fit a batch size of 2 on a 12GB GPU. We attempted to scale it up for the FGVC datasets, fitting a batch size of 16 across $4\times$ 24GB GPUs, with training taking a week. However, we found its performance after a week to be slightly lower than ARPL+ in this setting.

## E Comparisons with other deep learning based OSR methods

Table 6: **Comparing our improved baseline with other deep learning based OSR methods on the standard benchmark datasets.** All results indicate the area under the Receiver-Operator curve (AUROC) as a percentage. We also show the backbone architecture used for each method, showing results with multiple backbones when reported.

| Method | Backbone | MNIST | SVHN | CIFAR10 | CIFAR + 10 | CIFAR + 50 | TinyImageNet |
|---|---|---|---|---|---|---|---|
| MSP (Neal et al., 2018) | VGG32 | 97.8 | 88.6 | 67.7 | 81.6 | 80.5 | 57.7 |
| OpenMax (Bendale & Boult, 2016) | VGG32 | 98.1 | 89.4 | 69.5 | 81.7 | 79.6 | 57.6 |
| G-OpenMax (Ge et al., 2017) | VGG32 | 98.4 | 89.6 | 67.5 | 82.7 | 81.9 | 58.0 |
| OSRCI (Neal et al., 2018) | VGG32 | 98.8 | 91.0 | 69.9 | 83.8 | 82.7 | 58.6 |
| CROSR (Yoshihashi et al., 2019) | DHRNet | 99.1 | 89.9 | - | - | - | 58.9 |
| C2AE (Oza & Patel, 2019) | VGG32 | 98.9 | 92.2 | 89.5 | 95.5 | 93.7 | 74.8 |
| GFROSR (Perera et al., 2020) | VGG32 / WRN-28-10 | - | 93.5 / 95.5 | 80.7 / 83.1 | 92.8 / 91.5 | 92.6 / 91.3 | 60.8 / 64.7 |
| CGDL (Sun et al., 2021) | CPGM-AAE | 99.5 | 96.8 | **95.3** | 96.5 | 96.1 | 77.0 |
| OpenHybrid (Zhang et al., 2020) | VGG32 | 99.5 | 94.7 | 95.0 | 96.2 | 95.5 | 79.3 |
| RPL (Chen et al., 2020a) | VGG32 / WRN-40-4 | 99.3 / 99.6 | 95.1 / 96.8 | 86.1 / 90.1 | 85.6 / 97.6 | 85.0 / **96.8** | 70.2 / 80.9 |
| PROSER (Zhou et al., 2021) | WRN-28-10 | - | 94.3 | 89.1 | 96.0 | 85.3 | 69.3 |
| ARPL (Chen et al., 2021) | VGG32 | 99.6 | 96.3 | 90.1 | 96.5 | 94.3 | 76.2 |
| ARPL + CS (Chen et al., 2021) | VGG32 | **99.7** | 96.7 | 91.0 | 97.1 | 95.1 | 78.2 |
| OSRCI+ | VGG32 | 98.5 (-0.3) | 89.9 (-1.1) | 87.2 (**+17.3**) | 91.1 (**+7.3**) | 90.3 (**+7.6**) | 62.6 (**+4.0**) |
| (ARPL + CS)+ | VGG32 | 99.2 (-0.5) | 96.8 (**+0.1**) | 93.9 (**+2.9**) | **98.1** (**+1.0**) | **96.7** (**+1.6**) | 82.5 (**+4.3**) |
| Baseline (MSP+) | VGG32 | 98.6 (**+0.8**) | 96.0 (**+7.4**) | 90.1 (**+22.4**) | 95.6 (**+14.0**) | 94.0 (**+13.5**) | 82.7 (**+25.0**) |
| Baseline (MLS) | VGG32 | 99.3 (**+1.5**) | **97.1** (**+8.5**) | 93.6 (**+25.9**) | 97.9 (**+16.3**) | 96.5 (**+16.0**) | **83.0** (**+25.3**) |

In table 6, we provide comparisons with more methods, including those using a different backbone architecture, to supplement table 1 from the main paper. The overall conclusion is the same as in the main paper. Specifically, our improved baseline significantly outperforms reported baseline figures and outperforms state-of-the-art OSR models on a number of standard benchmarks. Training other OSR methods (OSRCI, ARPL + CS (Neal et al., 2018; Chen et al., 2021)) on top of our improved baseline can boost also their OSR performance. However, the discrepancy between the state-of-the-art and the baseline is now negligible.

## F Out-of-Distribution Detection Results

In this section, we run experiments on OoD benchmarks, a separate but related machine learning sub-field to OSR. OoD deals with all forms of distributional shifts, whereas OSR focusses on *semantic* novelty. Specifically, in the 'multiclass' OoD setting, a model is trained for classification on a given dataset, before being tasked with detecting test samples from other datasets as 'unknown' (Hendrycks & Gimpel, 2017). Once again, this task is evaluated as a binary classification ('known'/'unknown') problem. A notable difference with the OSR setting is that OoD models often have access to auxiliary data as examples of 'OoD' during training (Hendrycks et al., 2019).

### F.1 Correlation between closed-set and OoD performance

First, we conduct similar experiments to sec. 3. We evaluate four ResNet models trained on CI-FAR100 on the OoD task, using CIFAR10 for examples of 'OoD'. We show the closed-set and OoD performances of these models are correlated in fig. 8, with a Pearson Product-Moment correlation of $\rho = 0.97$. This trend is similar to the one observed in the ImageNet OSR evaluation in fig. 3b.

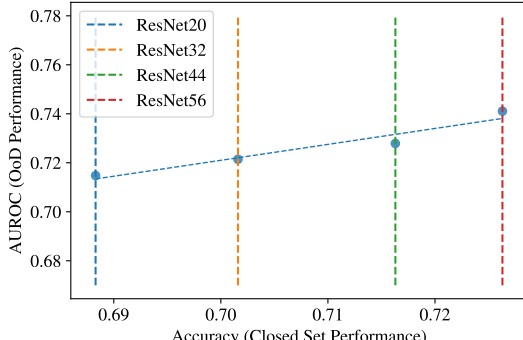

Figure 8: **OoD against closed-set performance for four ResNet models trained on CIFAR100, using CIFAR10 as OoD.** The plot indicates a similar performance correlation as observed in fig. 3b.

### F.2 OoD Performance With Differing Types of Distribution Shift

Next, in table 7, we evaluate OoD performance when different datasets are taken as examples of 'OoD' with respect to CIFAR100. Specifically, we compare OSR methods (and an OoD baseline), taking Gaussian Noise, SVHN and CIFAR10 as 'OoD'.

Table 7: **Results on out-of-distribution detection benchmarks.** We evaluate two MLS models: one represents a model which we train ourselves; the second represents a strong pre-trained model from (Lim et al., 2019).

|  | Outlier Exposure (Hendrycks et al., 2019) | OpenHybrid (Zhang et al., 2020) | ARPL+CS | MLS | MLS (Lim et al., 2019) |
|---|---|---|---|---|---|
| CIFAR100 → Gaussian Noise | **95.7** | - | 67.6 | 73.5 | 78.9 |
| CIFAR100 → SVHN | 86.9 | - | 77.9 | 83.3 | **88.9** |
| CIFAR100 → CIFAR10 | 75.7 | **85.6** | 73.0 | 77.7 | 83.2 |

As a strong baseline from the OoD literature, we report results from Outlier Exposure (O.E.) (Hendrycks et al., 2019), which encourages the classifier to predict a uniform distribution when fed auxiliary 'OoD' images from 80 Million Tiny Images (Torralba et al., 2008). We also report results from OpenHybrid (Zhang et al., 2020) which reports a CIFAR100 → CIFAR10 result. Furthermore, we train ARPL+CS and MLS in this setting, training a ResNet50 for 200 epochs. As a final experiment, we take a strong model pre-trained on CIFAR100 from (Lim et al., 2019) and evaluate it on the OoD benchmarks. Our results show that, while OpenHybrid performs strongly on the CIFAR100 → CIFAR10 experiment, the two MLS models outperform the O.E baseline on this evaluation *despite not having seen extra data during training*.

### F.3 Evaluation on OoD Benchmarks

Finally, we run our MLS method on the standard OoD benchmark suite. Specifically, we take models trained on CIFAR10 and CIFAR100, and evaluate them when Places365 (Zhou et al., 2017), Textures (Cimpoi et al., 2014), LSUN-Crop (Yu et al., 2015), LSUN-Resize (Yu et al., 2015), iSUN (Xu et al., 2015) and SVHN (Netzer et al., 2011) are used in turn as 'OoD' datasets. We take well-trained WideResNet-40 models (trained with Fast Auto-Augment on CIFAR10 and CIFAR100 from (Lim et al., 2019)) and run our MLS baseline on top. We compare against state-of-the-art OoD methods which do not use extra data for fine-tuning, and report our results in table 8. We report average AUROC across the six OoD datasets.

We find that strong closed-set classifiers with our MLS baseline can achieve highly competitive performance on the OoD benchmarks, once again substantially closing the gap between the MSP baseline (Hendrycks & Gimpel, 2017) and state-of-the-art.

Table 8: **Results of our strong baseline on the full OoD benchmark suite.** We take strong WideResNet-40 models from (Lim et al., 2019) and run our MLS baseline on top. Models are trained on CIFAR10 and CIFAR100 as 'in-distribution' and we report AUROC averaged across six OoD datasets. All compared figures are taken from (Du et al., 2022) and Liu et al. (2020).

| Method | CIFAR10 | CIFAR100 |
|---|---|---|
| MSP  (Hendrycks & Gimpel, 2017) | 90.9 | 75.5 |
| ODIN  (Liang et al., 2018) | 91.1 | 77.4 |
| Energy Score  (Liu et al., 2020) | 91.9 | 79.6 |
| Mahanabolis  (Lee et al., 2018b) | 93.3 | **84.1** |
| VOS  (Du et al., 2022) | 94.1 | - |
| MLS (Ours) | **95.1** | 80.8 |

**Discussion.** Our results show that strong closed-set classifiers can also perform well in the OoD setting, even compared to very recent methods such as Virtual Outlier Synthesis (VOS, (Du et al., 2022)). In fact, in some cases, we find the MLS baseline exceeds state-of-the-art for this task.

Interestingly, the MLS baseline performs best with in the 'near-OoD' case (*e.g.* SVHN and CIFAR10 as 'OoD' in table 7, *i.e.* in the more similar settings to OSR). In fact, the MLS models trained on CIFAR100 are *worse* at detecting Gaussian Noise than CIFAR10 images as 'OoD'. We present this peculiar finding as evidence that the OoD and OSR research questions may have different, and possibly orthogonal, solutions. We hope that benchmarks which can isolate *semantic novelty* from low-level distributional shifts, such as the Semantic Shift Benchmark from sec. 5, can facilitate more controlled OSR and OoD research.

## G  DISCUSSION: UNDERSTANDING SYSTEMS OF CATEGORIZATION

Before one can establish if an image belongs to a *new* class, one must first understand what constitutes a *single* class, or how the system of categorization is constructed. To illustrate this, consider a classifier trained on instances of two household pets: {*Labrador (dog), British Shorthair (cat)*}. Now consider an open-world setting in which the model must be able to distinguish previously unseen objects, perhaps: {*Poodle (dog), Sphynx (cat)*}. In this case, understanding the categorization system is essential to making the open-set decision. Does the classification system delineate individual animal species? In this case, both 'Poodle' and 'Sphynx' should be identified as 'open-set' examples. Or does it instead simply separate 'cats' from 'dogs'? In which case neither object belongs to the open-set.

To solve this problem, and to perform OSR reliably, the model must understand the set of invariances within a single category, as well as a set of 'axes of variation' to distinguish between categories. Specifically, different instances within a single category will have a set of features which can be freely varied without the category label changing. In computer vision, this often refers to characteristics such as pose and lighting, but could also refer to more abstract features such as animal gender or background setting. Meanwhile, the classification system will also have a (possibly abstract) set of axes of variation to which the category label is sensitive.

In the current OSR benchmarks, with either abstract class definitions or a small number of classes, the set of axes of variation which can distinguish between categories is diverse. In this sense, the problem is *ill-posed*, with many axes likely being equally valid to distinguish between the training classes, including those based on semantically meaningless low-level features. In contrast, within our proposed fine-grained setting, the set of axes of variation which can distinguish between categories is far more constrained. For instance, in the CUB case, given a training task of classifying 100 bird species, there is little uncertainty as to what the axis of semantic variation could be.

## H  CREATING SPLITS FOR THE SEMANTIC SHIFT BENCHMARK

### H.1  SPLIT CONSTRUCTION

In sec. 5 of the main paper, we sketched the process for constructing open-set splits from the CUB dataset. Here, we describe the process in detail for both CUB, Stanford Cars and FGVC-Aircraft, which each have different attribute structures.

For each FGVC benchmark, we split its classes into two disjoint sets, $\mathcal{C}$ and $\mathcal{U}$, containing closed-set and open-set classes respectively. $\mathcal{U}$ is further subdivided into disjoint {'Easy', 'Medium', 'Hard'} sets with varying degrees of attribute similarity with any class in $\mathcal{C}$. Specifically, we measure the difficulty of an open-set class by its semantic similarity with its *most similar* training class (where similarity is defined in terms of attribute overlap).

In practice, we found the semantic similarity of the 'Medium' and 'Hard' splits of Stanford Cars to the closed-set to be very similar, hence we combine them into a single 'Hard' split.

**CUB.** In CUB, each image is labelled for the presence of 312 visual attributes such as `has_bill_shape::hooked` and `has_breast_color::yellow`. Note that images from the same class do not all share the same attributes, both because of standard factors such as pose and occlusion, but also because of factors such as the age and gender of the bird.

This information is summarised on a per-class basis, describing how often each attribute occurs in each class; *i.e.*, a matrix $M \in [0,1]^{C \times A}$ is available, where $C = 200$ is the total number of classes in CUB and $A = 312$ is the number of attributes. This allows us to construct a class similarity matrix $S \in [0,1]^{C \times C}$ where $S_{ij} = \mathbf{m}_i \cdot \mathbf{m}_j$ and $\mathbf{m}_i$ is the L2-normalized $i^{th}$ row of $M$. Thus, given a set of closed-set classes in $\mathcal{C}$, we can rank all remaining classes ($\mathcal{U}$) according to their maximum similarity with any of the training classes. Finally, we bin the ranked open-set classes into {'Easy', 'Medium', 'Hard'} sets. In practice, we randomly sample 1 million combinations of $\mathcal{C}$, and select the combination which results in the most difficult open-set splits.

**Stanford Cars.** Each class name in Stanford Cars follows the format of 'Make'-'Model'-'Type'-'Year'; for instance 'Aston Martin - V8 Vantage - Convertible - 2012' is a class. In this case, we create open-set splits of different difficulties based on the similarity between class names.

We first create the 'Hard' open-set split by identifying pairs of classes which have the same 'Make', 'Model' and 'Type' but come from different 'Years'. Next, we create the 'Medium' split from class pairs which have the same 'Make' and 'Model' but have different 'Types'. Finally, the 'Easy' split is constructed from pairs which have the same 'Make' but different 'Models'.

We note that open-set bins of different difficulties in Stanford Cars are the most troublesome to define. This is because the rough hierarchy in the class names may not always correspond to the degree of visual similarity between the classes. For instance, two cars from the same 'Year' but of different 'Makes' (*e.g.* a Ford and Nissan both made in 2010) may look more similar than cars of the same 'Make'-'Model'-'Type' but from different years (*e.g.* Audi S4 Sedan 2007 and Audi S4 Sedan 2012).

**FGVC-Aircraft.** We leverage the hierarchy of class labels in FGVC-Aircraft; each image is labelled with a 'manufacturer' (*e.g.*, 'Airbus' or 'Boeing'), a 'family' (*e.g.*, 'A320' or 'A330') and a 'variant' (*e.g.*'A330-200' or 'A330-300'). The hierarchy is constructed as a tree, with 'manufacturer' classes at the top level, 'family' classes at the second, and 'variant' classes at the bottom. The standard image classification challenge operates at the variant level, meaning all variant classes are visually distinct with identifiable features. Furthermore, the hierarchy corresponds to visual similarity, i.e there is more inter-class variation between manufacturers than between variants from the same manufacturer. Thus, given the closed-set classes $\mathcal{C}$, we can create an 'Easy' open-set split from variants which do not share a manufacturer with any closed-set class. Meanwhile, 'Medium' open-set classes share a manufacturer with closed-set classes but come from different families, and 'Hard' open-set classes share families with closed-set classes but are different variants.

## H.2 SPLIT EXAMPLES

We include examples of images from the closed-set and open-set splits of the proposed FGVC datasets in fig. 9 and 11. For each dataset, we show examples of 'Easy' (green/top), 'Medium' (orange/middle) and 'Hard' (red/bottom) classes. For each difficulty, we show three images from three classes from the open-set (right) and their most similar class in the closed-set (left). We note that 'Hard' open-set classes are far more visually similar to their corresponding closed-set class than 'Easy' open-set classes.

## H.3 SPLIT DETAILS

All split details can be found here: https://github.com/sgvaze/osr_closed_set_all_you_need.

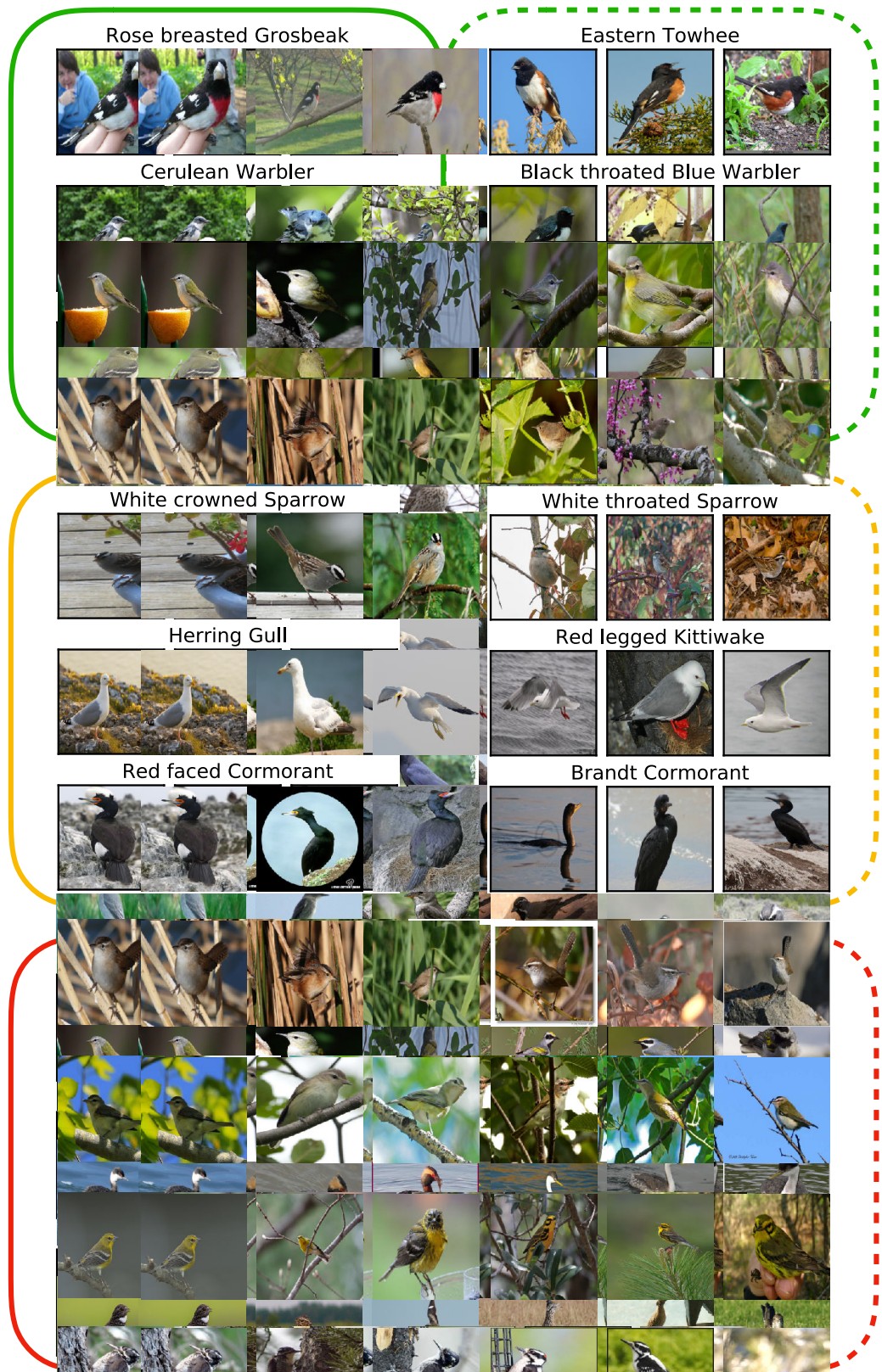

Figure 9: **Sample classes from closed and open-set splits for the CUB dataset.** We show 'Easy' (green/top), 'Medium' (orange/middle) and 'Hard' (red/bottom) classes. Classes on the left (solid outline) are in the closed-set, while classes on the right (dashed outline) are in the open-set.

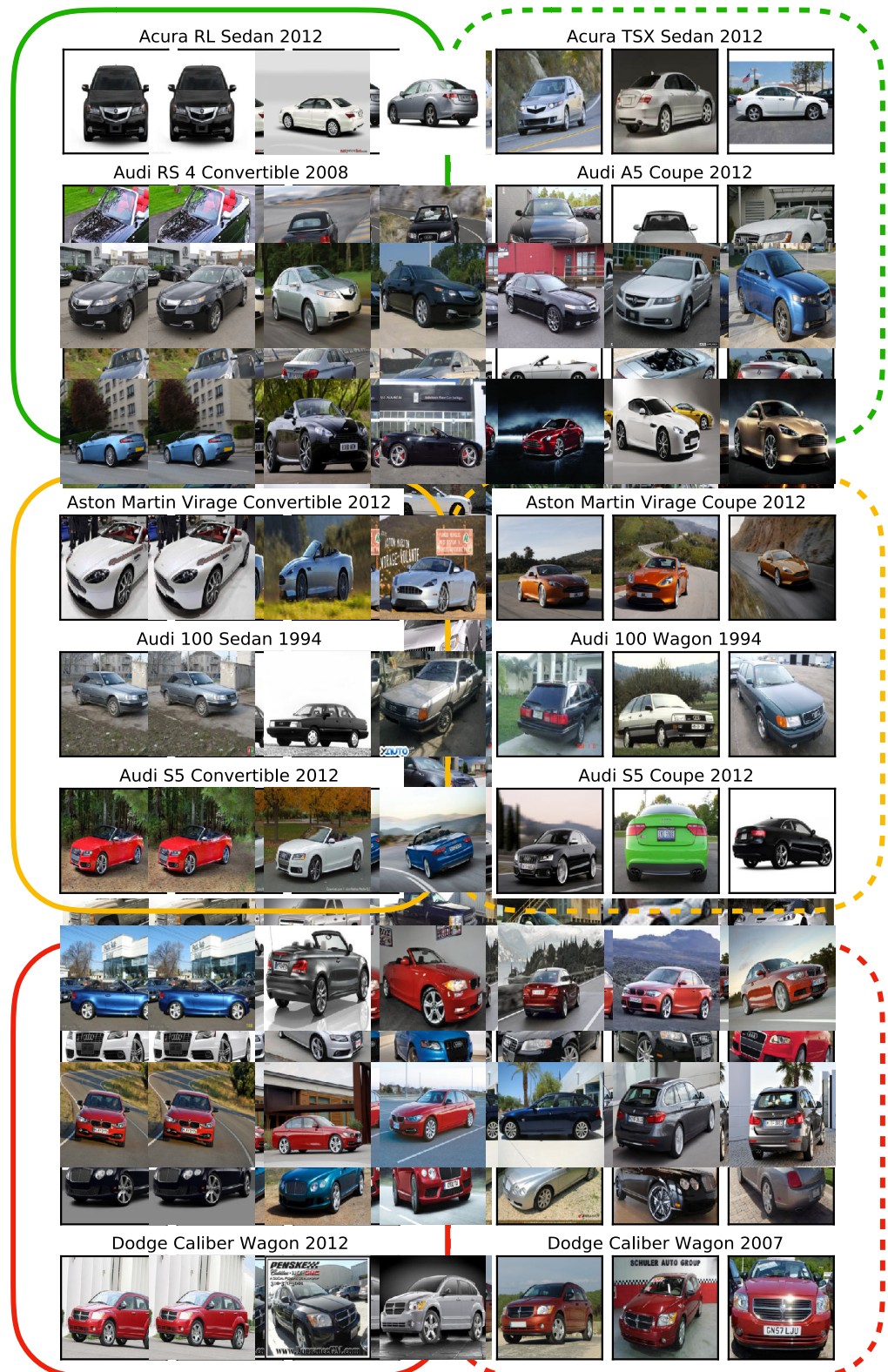

Figure 10: **Sample classes from closed and open-set splits for the Stanford Cars dataset.** We show 'Easy' (green/top), 'Medium' (orange/middle) and 'Hard' (red/bottom) classes. Classes on the left (solid outline) are in the closed-set, while classes on the right (dashed outline) are in the open-set. *In practice, we combine the 'Medium' and 'Hard' splits during evaluation.*

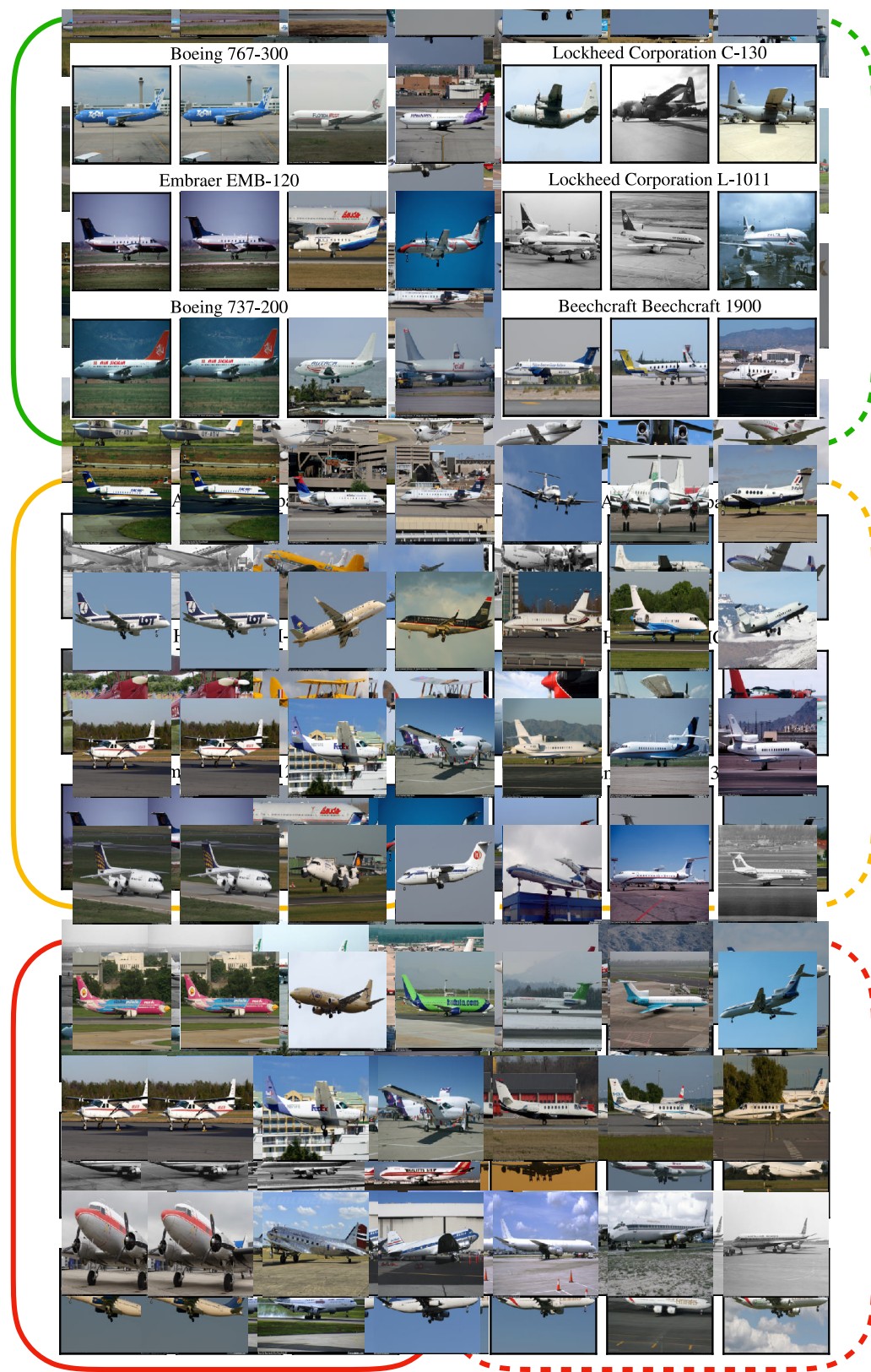

Figure 11: **Sample classes from closed and open-set splits for the FGVC-Aircraft dataset.** We show 'Easy' (green/top), 'Medium' (orange/middle) and 'Hard' (red/bottom) classes. Classes on the left (solid outline) are in the closed-set, while classes on the right (dashed outline) are in the open-set.

## I    AVERAGE PRECISION EVALUATION ON PROPOSED BENCHMARKS

We report average precision (AP) for the binary 'known/unknown' decision for the proposed benchmark evaluations in table 9. AP is a standard metric in the OoD literature and is better suited for dealing with class imbalance at test time. We note that the 'Hard' FGVC open-set splits (with a small number of classes) report substantially poorer AP than AUROC in absolute terms. We treat open-set examples as 'positive' during evaluation.

Table 9: **Average Precision (AP) results on the proposed benchmark datasets for 'Easy' / 'Medium' / 'Hard' splits.**

|       | CUB | FGVC-Aircraft | ImageNet |
|-------|-----|---------------|----------|
| ARPL+ | 59.9 / 53.3 / 45.3 | 66.9 / **58.9** / 34.4 | **78.2** / - / **71.2** |
| MLS   | **67.1 / 58.2 / 47.2** | **69.2** / 58.2 / **39.6** | 76.6 / - / 68.6 |

