# OpenReview forum: "Open-Set Recognition: A Good Closed-Set Classifier is All You Need"
_ICLR.cc/2022/Conference — ICLR 2022 Oral_

### Official Review · Reviewer_dw7J · 2021-11-02

**Correctness:** 4
**Technical Novelty And Significance:** 3
**Empirical Novelty And Significance:** 3
**Recommendation:** 8
**Confidence:** 4

**Main Review:**

Strengths:
- The paper is well written, clear and straight to the point.
- The appendix add more experiments and details to further understand the main paper statements.
- Up to my knowledge the most important references in the field have been considered
- The paper belongs to the category of very important papers that show that sometimes in research we consider very complex solutions, whereas a properly tuned standard approach can do the job as well.
- The authors propose a clear distinction between open-set recognition (in which the none-of-the above class comes form well defined classes) and out-of-distribution (in which the none-of-the-above can be any kind of image). It is important to clarify the different tasks.
- It is interesting to see that in new and not overfitted datasets, cross-entropy baseline seems to perform better than the best method for open-set scenario.

Weaknesses:
- Authors compare with only ARPL and OpenHybrid approaches because they say that the other approaches perform lower on open-set benchmarks. I think it is still important to consider also the other methods especially because the authors propose new benchmarks and on these benchmarks the ranking between the methods changes. So it could be for the other methods. I see that a more complete evaluation is performed in the appendix E (Table 6), however this is not done for the new datasets (Table 3).
- In Fig.2 I would expect to see also results for OpenHybrid. Also, in the figure, for the simplest datasets (with performance close to 100) there is not much correlation between open-set can close-set performance as ARPL does not improve over cross-entropy on the close-set scenario. It is something that should be mentioned.
- Results in Table 1 for the cross-entropy are presented with a ranking based on the classifier logits, which in my opinion makes more sense. However, as previous works use the softmax scores, it is important to compare the two ways. I see a comparison in Appendix B (Fig. 6c), but this is only for a model. I would like to see this comparison at least for the models in tab. 1.

Additional questions/comments:
- Fig.1a presents ARPL in the legend, without really having yet introduced the method.
- The authors propose a clear distinction between open-set recognition and out-of-distribution. However, is this distinction really necessary. Could not the two fields be unified? It would be more interesting to know if the same conclusions of this paper are valid also for out-of-distribution problems.
- The "more theoretical" justification of the results at bottom of page 4 does not add much, but I do not have suggestions on how to improve it.
- In Fig.3 it is interesting to see that ViT seems to have a better generalization to the open-set scenario on ImageNet. Is it due to the fact that it has been trained on more data or it is the reduced inductive bias (no convolutions)? It would be interesting to see if with different sizes of ViT, the correlation between open and close-set scenario still holds.

**Summary Of The Paper:**

This paper considers the problem of open-set recognition, in which a visual classifier should be able to distinguish images of the trained categories from images of other different categories.
The authors show that there is a strong correlation between results on open-set scenario and the close set-scenario (the classic problem in which the model is trained and tested on the same semantic categories). Then, they show that a baseline based on ranking the logits of a model trained on standard cross-entropy training can be very competitive with more complex methods when trained with strong data augmentation and other improvements.
Finally, they propose new benchmarks for open-set recognition emphasising the fact that in contrast to other similar tasks, in open set scenario, the images that should be classified as other classes not seen during training should belong to a different semantic category, and not to other kind of distributional shifts.
The take home message of this paper is that at this stage of research for improving open-set recognition, the best that we can do (or almost) is to use stronger models for close-set recognition (as the title suggests).

**Summary Of The Review:**

The paper is well written and introduces interesting results that can change the understanding of the open-set recognition. Experimental results prove the main conclusion of the paper, but some additional experiments are needed (see above) to further understanding the problem. Also, in my understanding the distinction between open-set recognition and out-of-distribution is ficticial and the proposed cross-entropy baseline should evaluated also on out-of-distribution settings.

---

> ### Author Response · Authors · 2021-11-22
> **Response to Reviewer dw7j (2/2)**
>
> >“In Fig. 2 I would expect to see also results for OpenHybrid”:
>
> We did not include OpenHybrid results in this figure for the same reasons as mentioned above (we could not recreate their results with our computational resources). However, we could include OSRCI results in this figure, as we found this model tractable to train on the old (smaller scale) OSR benchmarks. We thank the reviewer for this suggestion and have included this version of the figure in Appendix A (Fig. 6).
>
> >“In the figure [Fig. 2], for the simplest datasets (with performance close to 100) there is not much correlation between open-set can close-set performance”:
>
> This is an interesting point. It is indeed the case that the trend may behave differently when closed-set accuracies become very high or saturated. We have updated the manuscript to include this nuance in Appendix A.
>
> >“In Fig.3 it is interesting to see that ViT seems to have a better generalization to the open-set scenario on ImageNet. Is it due to the fact that it has been trained on more data or it is the reduced inductive bias (no convolutions)? It would be interesting to see if with different sizes of ViT, the correlation between open and close-set scenario still holds”:
>
> We agree that it would be interesting to further analyze the reasons why ViT models perform well on OSR. Similar experiments were conducted in [1] (as pointed out by Reviewer HAFU). It is worth noting that all models in Fig 3, including ViT, were trained on the same data (ImageNet-1K).

---

> > ### Comment · Reviewer_dw7J · 2021-11-29
> > **Final Evaluation**
> >
> > I read the other reviews and I think there is a clear agreement that the paper is well written, clarify some important points about open-set recognition and out of distribution and shows that well tuned baselines based on cross-entropy are often very close (or even better in certain cases) to the performance of more complex and adapted methods.
> > I also read authors answers to my questions/comments and in most of the cases they give a clear answer about the impossibility to add some additional experiments or where possible they added the additional evaluation.
> > Thus, I keep my positive evaluation of this work and I recommend it for publication.

---

> ### Author Response · Authors · 2021-11-23
> **Response to Reviewer dw7j (1/2)**
>
> We thank the reviewer for their kind comments on our paper and for succinctly summarising our take-home message: “at this stage of research for improving open-set recognition, the best that we can do (or almost) is to use stronger models for close-set recognition”. We also concur that it is important for the research community (and practitioners) to understand when simple algorithms work as well as complex architectures, and that other similar papers are important! We also appreciate the comment on the importance of clarifying the tasks of OSR and OoD, and refer to the general response (above) for comments on experiments in the OoD setting.
>
> >“In my understanding the distinction between open-set recognition and out-of-distribution is ficticial and the proposed cross-entropy baseline should evaluated also on out-of-distribution settings”:
>
> We provided experiments in the OoD setting in Appendix F. We have now further augmented this section with results on the open-set/closed-set correlation in an OoD setting (please refer to the global response for details). To summarize, we found that:
>  - Taking a strong closed-set classifier on CIFAR100 can outperform a popular OoD baseline (Outlier Exposure) on common OoD evaluations.
>  - We could identify a similar correlation to Fig 3b in the main paper on the CIFAR100 → CIFAR10 OoD benchmark. We found a strong correlation between the closed-set accuracy of ResNets trained on C100, and their OoD performance on C10.
>
> *Regarding the OoD vs OSR distinction more broadly:* we agree with the reviewer that there is currently significant overlap in the research question tackled by the two fields. Hence we propose specific OSR benchmarks in Sec. 5, which isolate semantic novelty from general distribution shifts. In theory, we believe that OSR (semantic shift, e.g image of a bird → image of a different species of bird) is a subset of OoD (any distribution shift, including e.g image of a bird → Gaussian noise). As such, it may be the case that there are algorithms which work specifically well for the OSR problem; for instance those based on class-prototypes or class attributes. In fact, our results in Appendix F (Tab. 7) give some evidence that this is the case. This difference could be considered analogous to, for example, Fine-grained Image Recognition (CUB, FGVC-Aircraft) vs. General Image Recognition (ImageNet, CIFAR). We hope the proposed benchmarks will help study this distinction.
>
> >“Results in Table 1 for the cross-entropy are presented with a ranking based on the classifier logits  … However, as previous works use the softmax scores ... I would like to see this comparison at least for the models in Tab. 1”:
>
> Overall, due to their open-set scoring mechanisms, the other models in Tab. 1 are not suitable for performing an ablation on the ‘Softmax’/‘Logits’ scoring rule. Specifically:
>  - ARPL + CS uses the maximum distance with a set of learned ‘reciprocal points’ in feature space to provide the open-set score. We note that this could be seen as similar to logits (if one considers the reciprocal points as weights of a linear classifier).
>  - OpenHybrid does not use the classification vector for OSR scoring at all, but rather directly uses the scalar output of a density model as an open-set score.
>  - With OSRCI, the open-set score is computed by appending ‘0’ to the logit vector and normalizing with Softmax, thus creating a dummy probability for a ‘K+1’ class.
>
> >“It is still important to consider also the [performance of] other methods especially because the authors propose new benchmarks and on these benchmarks the ranking between the methods changes … I see that a more complete evaluation is performed in the appendix E (Table 6), however this is not done for the new datasets (Table 3)”:
>
> Overall, we found methods from Tab. 1, other than ARPL, to be too computationally expensive for us to train in the new setting in Tab. 3. We hope that, given that ARPL is a SOTA model in OSR research and the highly competitive performance of Cross-Entropy, our experiments will serve as sufficient baselines for future research. Specifically:
>
>  - We attempted to implement OpenHyrbid (after requesting code from the authors) but underperformed the reported numbers. We believe this to be due to the very large batch size (1024 in theirs vs 128 in ours) and large number of epochs (10k in theirs vs 600 in ours) required to stabilize the model.
>  - OSRCI requires training: a classifier and GAN concurrently; generating an auxiliary dataset with the GAN; and then training another classifier. Thus, it is expensive to train on large-scale datasets like ImageNet under a constrained hardware budget. Given the clarity of the ranking in Tab.1, we believe it still holds true in our new setting.

---

### Official Review · Reviewer_PPFy · 2021-11-02

**Correctness:** 4
**Technical Novelty And Significance:** 2
**Empirical Novelty And Significance:** 3
**Recommendation:** 6
**Confidence:** 4

**Main Review:**

This paper is dealing with the open set recognition problem which is a challenging research problem. Based on some findings about the correlation of closed set recognizer and open set recognizer, this paper found a way to achieve a strong OSR through an enhanced close set recognizer. This is a valuable finding for future open set recognition research.
The weakness of this paper is that there are not more in depth analysis the behind reason for these findings, thus the value of this paper is not that significant. So this is possibly a direction to make this paper even stronger.

**Summary Of The Paper:**

This paper first analyzed the correlation between closed set classifier and open set classifier. Then based on this finding, cross entropy closed set classifier is enhanced with a few recent accuracy improvement methods, then it is transferred to a open set classifier and achieve good performance. And also experimentation is conducted to compare with other SOTA close set classifiers, and that the proposed enhanced OSR classifier based on cross entropy baseline can achieve similar accuracy. Finally a new benchmark dataset is generated.

**Summary Of The Review:**

This paper proposed interesting finding about the correlation between closed set classifier and open set classifier, and leveraging this finding, a new SOTA open set classifier is generated from a baseline classifier, which has similar performance as  other SOTA close set classifiers. Also a new benchmark dataset is generated.
Overall it is a valuable paper, which is clearly written, and with good experimentation results to support. The weakness of this paper is lacking of more in depth analysis, and providing more insights to the behind rationale, which lower the value of this paper.
So overall I will recommend marginal acceptance of this paper.

---

> ### Author Response · Authors · 2021-11-22
> **Response to Reviewer PPFy**
>
> We thank the reviewer for the time they spent on our work, and for commenting on the value and clarity of the paper!
>
> >“there are not more in depth analysis the behind reason for these findings [closed-set / open-set correlation]”:
>
> This point was also raised by Reviewer ‘HAFU’, hence we address it in the general response (above). To summarize, we provide some intuitions for the empirical correlation in Sec 3.1 ‘Discussion’, as well as more in-depth analysis in Appendix B by looking at feature visualizations of the cross-entropy baseline.

---

> > ### Comment · Reviewer_PPFy · 2021-11-30
> > **Final evaluation**
> >
> > The updates by the authors addressed my concerns well, also consider the comments from other reviewers, I will keep my previous rating of this paper and suggest to accept this paper.

---

### Official Review · Reviewer_HAFU · 2021-11-07

**Correctness:** 4
**Technical Novelty And Significance:** 2
**Empirical Novelty And Significance:** 3
**Recommendation:** 8
**Confidence:** 4

**Main Review:**

The core technical contribution or claim of this paper is that closed set accuracy is important for openest detection. This is known for researchers working on openest detection, though not highlighted enough. For any representation one key challenge of open set detection is in distinguishing confusing instances (incorrect predictions) within closed set vs novel category instances along the decision boundary. So as the quality of closed set classification improves this overlap could decrease and hence would enable easier detection of open-set instances.

[1] makes similar claims that vision transformers result in better open-set detection without complex detection mechanisms. [3],[4] highlight the fact that good representation is all you need for meta learning or out of distribution generalization which though orthogonal to open-set detection, informs the research community that ‘quality of representation & performance on closed set’ is useful for auxiliary and relevant tasks of OOD, few-shot learning.

Similar to the observation of this paper of closed set accuracy, many works in ‘openworld’ competition [5] at CVPR has taken various strategies to improve closed set accuracy in pre-training phase to improve closed set accuracy and leverage it for open-set performance boost.
Another key contribution of this paper is proposing large scale ImageNet benchmark, [5] is a CVPR’21 competition and existing benchmark along the similar lines which unfortunately makes benchmarking contribution not significant enough.

Strengths:

This paper focuses on an important problem and highlights an observation and it’s simplicity is especially relevant for practical settings and safety-critical applications.
This paper is well written and has great empirical evaluation methodology and demonstrate SOTA compared to other methods with additional components.
This paper proposes additional benchmarks for openset detection leveraging fine-grained classification datasets, and this is adds a valuable dimension to openset benchmarks.

Weakness:

The core technical contribution(s) does not seem convincing to warranty a conference acceptance, especially given previous works making similar conclusions and contributions.
This lacks sufficient commentary on why a good closed set accuracy would result in better open-set detection.


Suggestion to Authors:
        Visualization of representations (albeit noisy) with and without augmentations/modified training could be insightful. May be including few examples of how hard novel instances/samples map onto different representations.
	For completeness please consider including CIFAR100 (close set) vs CIFAR10(open set), as the number of closed set classes increase the task of openset detection becomes more challenging, this version of experiment is included in [2] which this paper includes as baseline and [1].

References:

[1]  Exploring the Limits of Out-of-Distribution Detection. http://www.gatsby.ucl.ac.uk/~balaji/udl2021/accepted-papers/UDL2021-paper-001.pdf

[2] H. Zhang et al. Hybrid Models for Open Set Recognition https://www.ecva.net/papers/eccv_2020/papers_ECCV/papers/123480103.pdf

[3] Y. Tian et al. Rethinking Few-shot Image Classification: A Good Embedding is All You Need?
https://www.ecva.net/papers/eccv_2020/papers_ECCV/papers/123590256.pdf

[4] J. Miller et al. Accuracy on the Line: On the Strong Correlation Between Out-of-Distribution and In-Distribution Generalization. https://proceedings.mlr.press/v139/miller21b/miller21b.pdf

[5] Open World Vision CVPR’21 Competition and workshop https://www.cs.cmu.edu/~shuk/open-world-vision.html, https://eval.ai/web/challenges/challenge-page/1041/overview


**Summary Of The Paper:**

This paper makes an observation that good representations for open set detection would be correlated with high closed set accuracy.
To validate and illustrate this observation they improve closed set accuracy on existing open set benchmarks with different architectures and demonstrate  improvement in open set detection performance.
This paper also proposes new benchmarks for openest detection with fine grain details, which haven’t been studied in the setting of openest detection.

**Summary Of The Review:**

This paper focuses on an important problem of open-set detection and demonstrate that a good closed-set representation is a very strong baseline and competitive with more complex methods for open-set detection, which is valuable for many practical settings.
Though empirical evaluation is comprehensive and illustrate performance on large scale benchmarks, given existing prior works novelty of contributions seems rather limited

---

> ### Author Response · Authors · 2021-11-22
> **Response to Reviewer HAFU (2/2 + citations)**
>
> >“Another key contribution of this paper is proposing large scale ImageNet benchmark, [6] is a CVPR’21 competition and existing benchmark along the similar lines”:
>
> Previous works [6][7] have constructed open-set splits from the larger ImageNet database (we had cited [7] in Sec. 1 and Sec 5.1). Our contribution is to carefully structure the open-set classes for semantic similarity with the (closed-set) ImageNet-1K classes, thereby creating OSR splits of varying difficulty and allowing better understanding of the open-set problem. Our experiments suggest that this structuring is important. Specifically, we note that:
>  - Randomly sampling 10x more open-set classes from the larger ImageNet database reduces open-set performance by only 0.6%.
>  - In contrast, with our method, choosing open-set classes which are semantically more similar to ImageNet-1K (while keeping the number of open-set classes the same), reduces open-set performance by 6% (a 10x difference).
>  - We also propose two other benchmarks, based on FGVC datasets, which isolate semantic novelty from other low-level distributional shifts. In this case, harder splits reduce AUROC by as much as 19%.
>
> These observations are highlighted in Sec 5.2 ‘Results’ in our manuscript. We thank the reviewer for pointing out [6] and we have now cited it in our manuscript (Sec 5.1 ‘ImageNet for open-set recognition’).
>
> [1] D. Zhou et al.: Learning Placeholders for Open-Set Recognition \
> [2] R. Yoshihashi et al.: Classification-Reconstruction Learning for Open-Set Recognition \
> [3] Y. Tian et al.: Rethinking Few-shot Image Classification: A Good Embedding is All You Need? \
> [4] J. Miller et al.: Accuracy on the Line: On the Strong Correlation Between Out-of-Distribution and In-Distribution Generalization. \
> [5] Fort et al.: Exploring the Limits of Out-of-Distribution Detection \
> [6] Open World Vision CVPR’21 Competition and workshop \
> [7] Bendale & Boult: Towards Open Set Deep Networks

---

> ### Author Response · Authors · 2021-11-22
> **Response to Reviewer HAFU (1/2)**
>
> We thank the reviewer for their detailed feedback on our paper, and appreciate their acknowledgement of the practical utility of our findings (that the simple baseline performs comparably to SOTA), as well as our comprehensive experimental results. We refer the reviewer to the general response (top) for comments on CIFAR100 → CIFAR10 experiments and analysis on the closed-set/open-set correlation. Note: Analysis of the correlation is provided mainly in Appendix B, where we further provide feature space visualizations of weak / strong classifiers.
>
> >“[closed-set / open-set correlation] is known for researchers working on open-set detection, though not highlighted enough”:
>
> Although this correlation may be known to some researchers, we noted in Sec. 4 that increasingly sophisticated methods are proposed for OSR with carefully tuned training strategies and hyper-parameters. Meanwhile, the closed-set accuracy of the methods is often unreported in the literature, making it difficult to delineate what proportion of the open-set performance gains come from increases in closed-set accuracy. We thus perform controlled experiments to explore the correlation and further use our findings to demonstrate SOTA (or almost) open-set results with the simple cross-entropy baseline. We further note that:
>  - To our knowledge, few papers comment on both the closed-set and open-set performance. In fact, when the two are mentioned together, it is often highlighted that good OSR performance is achieved ‘without harming’ the closed-set accuracy [1][2].
>  - Though some methods proposed in the CVPR 2021 ‘Open World Vision’ workshop (mentioned in the review) discuss methods to improve the backbone, they do not rigorously explore how the improved accuracy affects the OSR performance.
>
> >“[3],[4] highlight the fact that good representation is all you need for meta learning or out of distribution generalization which though orthogonal to open-set detection, informs the research community that ‘quality of representation & performance on closed set’ is useful”:
>
> We believe our findings are different and complementary to those from [3] and [4].
>
> Specifically, [3] does not show that **better** closed-set representations transfer better to the few-shot task. Instead, they demonstrate that closed-set representations are good enough to provide few-shot features compared with bespoke few-shot algorithms. They also demonstrate that good self-supervised features work almost as well. In contrast, we specifically demonstrate that the better the closed-set representation, the better the open-set performance (i.e we demonstrate an explicit correlation between the two metrics).
>
> Secondly, [4] show that the better features can discriminate between a set of classes, the better those features can discriminate between the *same classes* under a distribution shift. This makes sense as these features have learned more invariances to spurious factors which benefit both in-distribution and out-of-distribution classification performance. In contrast, we show that these same features - which have learned more invariance to spurious factors - are better at detecting completely new classes.
>
> >“[5] makes similar claims that vision transformers result in better open-set detection":
>
> [5] shows that vision transformers outperform convolution models on an OoD benchmark. However we further show that ViTs not only give better open-set detection results, but also outperform the closed-set / open-set trend. Furthermore, this observation is not one of our key contributions, but an incidental finding amongst a larger study of the closed-set / open-set correlation over a wide range of architectures.
>
> Thank you for bringing this paper to our attention, we have now acknowledged this work in our manuscript (Sec 3.2, ‘Discussion’).

---

> > ### Comment · Reviewer_HAFU · 2021-11-26
> > **Updating rating to '8'**
> >
> > I appreciate author’s detailed feedback and updates to paper on various issues.
> >
> > I have updated my rating to ‘8’, I was concerned for novelty but contribution of paper is very useful for both researchers and practitioners. As pointed by reviewer 'dw7J' it is important to highlight simplicity. The take home message of this paper is very useful, that by leveraging augmentations, data, architectures (ViTs) if we improve the quality of representation on closed set then it reduces the gap between complex open-set detection mechanisms or even outperform them. But when the closed set accuracy is poor may be more complex methods compensate for quality of representation.
> >
> > I think this work is another evidence that a good representation (closed set performance, invariant/robust, etc.) is the crux of open set detection, few-shot learning, generalization to novel domains, similar to [3],[4] and other works.
> >
> > One additional comment would be, if the visualization plot can be updated from CIFAR10 to CIFAR100 (closed set) or ImageNet i.e. when the number of closed set classes are large.

---

### Author Response · Authors · 2021-11-22
**General Response**

We thank all reviewers for the constructive comments on our work. We appreciate the reviewers acknowledging the practical value of our findings for the research field and commenting on the comprehensive experimental evaluations. We found two comments which were common amongst more than one reviewer, hence we highlight them here.

>“lacks sufficient commentary on why a good closed set accuracy would result in better open-set detection”[HAFU] \
“there are not more in depth analysis the behind reason for these findings [open-set/closed-set correlation]”[PPFy]:

We included commentary on the closed-set/open-set correlation in the ‘Discussion’ of Sec 3.1 of the main paper and, particularly, in Appendix B. Specifically, we suggest that:
 - Taking results from the model robustness literature, classifiers which have a lower generalization error are likely to be better calibrated and hence be better OSR detectors (Sec 3.1).
 - Stronger cross-entropy classifiers map ‘seen’ class points further from the origin than weak classifiers, while mapping ‘unseen’ points near the origin. Thus, stronger closed-set classifiers have a stronger signal for the OSR decision (Appendix B).

>“Please consider including CIFAR100 (close set) vs CIFAR10 (open set)”[HAFU] \
“the proposed cross-entropy baseline should evaluated also on out-of-distribution settings”[dw7j]:

Experiments on an OoD benchmark were included in Appendix F (specifically, training on CIFAR100 as ‘in-distribution’ and testing on other datasets as ‘out-of-distribution’). We found that taking a strong closed-set classifier trained on CIFAR100 could outperform a common OoD baseline (Outlier Exposure). In general, however, it is difficult to perform comparisons between the fields as (unlike OSR) OoD allows access to extra data as examples of ‘OoD’ during training. We highlight this in our Related Work section (Sec. 2).

We have also expanded on this analysis in the updated manuscript. Notably, we have conducted further experiments in the OoD setting, conducting similar tests to those in Fig. 3b of the paper. Specifically, on the CIFAR100 → CIFAR10 OoD experiment, we find a correlation between the accuracy and OoD performance of a number of ResNet models. For ResNet-(20,32,44,56), we find a Pearson correlation coefficient of 0.97 between their closed-set accuracy and OoD detection performance. This plot is now included in Appendix F (Fig 8).

---

### Decision · Program_Chairs · 2022-01-20

**Decision:**

Accept (Oral)

**Comment:**

This paper provides well-written and thorough analysis demonstrating that closed-set recognition performance correlates with open-set recognition performance, and that simply making the close-set model strong via augmentation, label smoothing, etc. along with small scoring changes (using logits rather than softmax probabilities) can get close to (or better than in some cases) performance than much more complicated methods. The authors also propose a large-scale benchmark that varies the semantic similarity across classes, allowing for a more fine-grained analysis of this problem.

Overall, all of the reviewers thoughts that the paper provides very thorough validation of an insight that would be very interesting to the community. Reviewer HAFU had some concerns about novelty, since a number of papers have shown closed-set classifier improvements (and therefore better embeddings) benefit related problems such as few-shot learning and generalization to novel domains, as well as proposed large-scale experiments. The rebuttal convinced this reviewer, however, that some of the contributions and findings are unique and provide additional evidence to the community, and the new setting provides more fine-grained analysis. Reviewer dw7J had a number of suggestions in terms of additional evaluations, and the rebuttal either clarified why it is not possible or added them. As a result, after the discussion the reviewers all supported acceptance of this paper.

Given the above discussion, and rebuttal/changes to the paper, I recommend acceptance. It is a very well-done empirical paper, provides interesting findings, stronger baselines, and thorough experimentation. Further, some of the smaller findings (ViT correlation experiment) as well as larger relationship between open-set recognition and out-of-distribution detection are valuable contributions to the community. Finally, I would recommend this paper as oral, given that it may garner a good discussion of these contributions.

---

> ### Public Comment · ~Glen_Fowler1 · 2022-12-19
> **Thanks**
>
> Thanks for the info in brief.